# Attenuation of PKCδ enhances metabolic activity and promotes expansion of blood progenitors

Tata Nageswara Rao[1,2,*,†] (iD), Manoj K Gupta[2], Samir Softic[3,4], Leo D Wang[1,2,5,‡], Young C Jang[1,2,§], Thomas Thomou[3], Olivier Bezy[3,¶] (iD), Rohit N Kulkarni[2], C Ronald Kahn[3] (iD) & Amy J Wagers[1,2,**] (iD)

## Abstract

A finely tuned balance of self-renewal, differentiation, proliferation, and survival governs the pool size and regenerative capacity of blood-forming hematopoietic stem and progenitor cells (HSPCs). Here, we report that protein kinase C delta (PKCδ) is a critical regulator of adult HSPC number and function that couples the proliferative and metabolic activities of HSPCs. PKCδ-deficient mice showed a pronounced increase in HSPC numbers, increased competence in reconstituting lethally irradiated recipients, enhanced long-term competitive advantage in serial transplantation studies, and an augmented HSPC recovery during stress. PKCδ-deficient HSPCs also showed accelerated proliferation and reduced apoptosis, but did not exhaust in serial transplant assays or induce leukemia. Using inducible knockout and transplantation models, we further found that PKCδ acts in a hematopoietic cell-intrinsic manner to restrict HSPC number and bone marrow regenerative function. Mechanistically, PKCδ regulates HSPC energy metabolism and coordinately governs multiple regulators within signaling pathways implicated in HSPC homeostasis. Together, these data identify PKCδ as a critical regulator of HSPC signaling and metabolism that acts to limit HSPC expansion in response to physiological and regenerative demands.

**Keywords** hematopoietic stem and progenitors; metabolism; PKCδ; regeneration; signaling
**Subject Categories** Immunology; Metabolism; Stem Cells
**The EMBO Journal (2018) 37: e100409**

## Introduction

Hematopoietic stem cells (HSCs) are a specialized subset of cells equipped with two cardinal features, self-renewal and multi-lineage differentiation potency, which are critical for their function in bone marrow transplantation (Orkin & Zon, 2008). The regulation of HSC self-renewal and differentiation in balance with proliferation and apoptosis governs the size and functional capacity of the stem cell pool, while defects in such regulation can result in leukemic transformation or depletion of normal hematopoietic activity (Yilmaz *et al*, 2006; Pietras *et al*, 2011; Wagers, 2012). Recent studies have revealed multiple important HSC regulators and suggest that an orchestrated interaction among cell-intrinsic signals (transcription factors, cell surface receptors, cell cycle regulators, and signal transducers) and extrinsic mediators (bone marrow niche components and soluble growth factors) controls HSC fate (Orkin & Zon, 2008; Wilson *et al*, 2009; Ehninger & Trumpp, 2011; Gazit *et al*, 2013). Furthermore, growing evidence suggests a link between the metabolic activity of HSCs and their capacity to preserve stem cell functions and hematopoietic differentiation potential (Yu *et al*, 2013; Burgess *et al*, 2014; Kohli & Passegue, 2014). Despite these advances, however, many of the molecular pathways and mechanisms that regulate the self-renewal, survival, expansion, and regenerative functions of blood-forming stem cells remain unknown. Improved understanding of the molecular machinery that determines HSC function will aid the development of innovative strategies to expand HSCs *ex vivo* and to prevent their involvement in hematopoietic cancers.

Protein kinase δ (*PKCδ*) is a member of the novel subclass of PKC serine/threonine kinase isoforms and has been implicated in regulating key cellular processes including proliferation, apoptosis, differentiation, and metabolism via its role in diverse downstream signal transduction pathways (Basu & Pal, 2010; Bezy *et al*, 2011).

1  Department of Stem Cell and Regenerative Biology, Harvard Stem Cell Institute, Harvard University, Cambridge, MA, USA
2  Section on Islet Cell and Regenerative Biology, Joslin Diabetes Center, Boston, MA, USA
3  Section on Integrative Physiology and Metabolism, Joslin Diabetes Center, Boston, MA, USA
4  Division of Gastroenterology, Hepatology and Nutrition, Boston Children's Hospital, Boston, MA, USA
5  Division of Pediatric Hematology/Oncology/Stem Cell Transplantation, Dana-Farber/Boston Children's Center for Cancer and Blood Disorders, Boston, MA, USA
*Corresponding author. Tel: +41 768030539; E-mail: rao.tata@unibas.ch
**Corresponding author. Tel: +1 6174960586; E-mail: amy_wagers@harvard.edu
†Present address: Department of Biomedicine, Experimental Hematology, University Hospital Basel and University of Basel, Basel, Switzerland
‡Present address: Departments of Immunooncology and Pediatrics, Beckman Research Institute, City of Hope, Duarte, CA, USA
§Present address: Georgia Institute of Technology, School of Biological Sciences, Parker H. Petit Institute for Bioengineering and Bioscience, Atlanta, GA, USA
¶Present address: Internal Medicine Research Unit, Pfizer Inc., Cambridge, MA, USA

Although the role of *PKCδ* in apoptosis appears to be stimulus- and context-dependent, in most cases, overexpression or activation of *PKCδ* induces apoptosis (Basu & Pal, 2010). PKCδ can be activated by diacyl glycerol (DAG) and phorbol esters (such as PMA) (Basu & Pal, 2010), which triggers a pro-apoptotic signaling cascade that may include proteolytic activation and translocation of PKCδ to the mitochondria (Limnander *et al*, 2011).

PKCδ is widely expressed in the mouse and human hematopoietic systems (Limnander *et al*, 2011), and one of its best-studied functions in hematopoiesis is in B-cell signaling, where PKCδ deficiency enhances B-cell proliferation and leads to autoimmunity in mice and man (Miyamoto *et al*, 2002; Guo *et al*, 2004; Limnander *et al*, 2011). However, the role of PKCδ in regulating more primitive hematopoietic precursors, including blood-forming hematopoietic stem and progenitor cells (HSPCs), has not been interrogated. Given the established role of PKCδ in regulating cell survival, proliferation, and metabolism (Basu & Pal, 2010; Bezy *et al*, 2011), and the fact that perturbations in cell survival, proliferation, or metabolic signals can disrupt the functional integrity of HSPCs, we hypothesized that PKCδ may play a role in regulating HSPC homeostasis. Here, we test this hypothesis using *in vivo* and *in vitro* approaches and demonstrate that PKCδ restricts HSPC number and function in the steady-state and during hematopoietic stress conditions. $PKC\delta^{-/-}$ mice showed increased numbers of HSPCs in the bone marrow and better competence in bone marrow transplantation assays. Mechanistically, these phenotypes could be attributed to accelerated cell cycle progression and reduced apoptosis in *PKCδ*-deficient HSPCs. Finally, we found that *PKCδ*-deficient HSPCs display enhanced mitochondrial activity and oxidative phosphorylation. Our results indicate a pivotal role for PKCδ in the regulation of HSPC proliferation and apoptosis and identify PKCδ as a critical rheostat controlling HSPC expansion. Thus, targeting of PKCδ may represent an attractive strategy to stimulate *ex vivo* expansion of HSPCs and enhance hematopoietic recovery following HSPC transplantation.

## Results

### PKCδ deficiency expands the primitive HSC pool *in vivo*

To investigate the role of PKCδ in hematopoietic homeostasis, we first examined its expression during steady-state blood cell differentiation in the bone marrow of wild-type mice. Real-time PCR analysis of PKCδ mRNA in FACS-sorted Lineage negative (Lin⁻), $Lin^-Sca1^+c\text{-}Kit^+$ (LSK) cells, LT-HSC, ST-HSC, MPP, myeloid progenitors (GMP, CMP, MEP), and common lymphoid progenitor (CLP) subsets revealed that *PKCδ* is expressed at variable levels by all HSPC populations, with the highest expression in CLP, LT-HSC, and MPPs. The lowest levels of PKCδ expression were observed in megakaryocyte-erythroid progenitors (MEP) (Fig 1A). This expression pattern suggests that PKCδ functions in primitive LT-HSCs, as well as in multiple other stages of hematopoiesis.

As an initial approach to probe the importance of PKCδ signaling in bone marrow (BM) HSPCs, we exposed FACS-purified wild-type HSPCs to Indolactam V (Indo-V), a non-specific PKC isoform activator, or Mallotoxin (MTX, also called Rottlerin), a PKC inhibitor that shows some selectivity for PKCδ when used at lower concentrations

(3–5 µM) (Gschwendt *et al*, 1994). Interestingly, Indo-V accelerated acquisition of lineage markers and depletion of primitive LSK HSPCs from the culture, whereas MTX increased the relative proportion of Lin⁻ cells and LSK cells (Fig EV1A–C). Consistent with enhanced maintenance of more primitive HSPCs in PKC-inhibited cultures, transplantation of MTX-treated HSPCs resulted in 2- to 3-fold higher contributions to mature hematopoietic lineages than transplantation of untreated control cells (Fig EV1D), although levels of engraftment were generally low in both experimental groups. These data suggest that the PKC pathway plays a role in regulating the number and/or function of hematopoietic reconstituting cells. To evaluate this possibility more specifically, we next determined the frequencies and absolute numbers of BM HSPCs in germline PKCδ knockout mice (Bezy *et al*, 2011), which allow for highly specific inactivation of PKCδ *in vivo*. PKCδ knockout mice displayed a slight but significant increase in BM cellularity (Appendix Fig S1A), and immunophenotypic analysis further revealed an increase in the frequency and absolute numbers (~2- to 3-fold increase) of LSK cells in the BM of *PKCδ*-deficient mice (Fig 1B, left plots). This expansion of LSK cells was unique to the *PKCδ*-deficient BM, as analysis of e14.5 fetal livers revealed equivalent frequencies of LSK cells in $PKC\delta^{-/-}$, $PKC^{-/-}$, and $PKC\delta^{+/+}$ mice (Appendix Fig S1B). More comprehensive analysis of the adult LSK compartment using Flt3- and CD34-based immunophenotypic fractionation indicated a significant increase in the frequency and absolute numbers of LT-HSC (CD34⁻Flt3⁻LSK), ST-HSC (CD34⁻Flt3⁻LSK), and MPP (CD34⁺Flt3⁺LSK) (Fig 1B and C). Consistent with this, "SLAM code" based analysis (Kiel *et al*, 2005) of BM LSK cells confirmed an ~3-fold increase in the frequency and numbers of LT-HSC (CD150⁺CD48⁻LSK) in PKCδ-deficient mice compared with wild-type littermates (Fig 1D).

Intriguingly, despite an increased frequency of primitive HSPCs in the BM of $PKC\delta^{-/-}$ mice, analysis of myeloid progenitors (Lin⁻Sca1⁻kit⁺, MyP) and further fractionation of these heterogeneous cell types into GMP (Lin⁻Sca1⁻kit⁺CD34⁺FcɛRγIII/II⁺), CMP (Lin⁻Sca1⁻kit⁺CD34⁺FcɛRγIII/Il^lo), and MEP (Lin⁻Sca1⁻kit⁺CD34⁺ FcɛRγIII/II⁺) revealed that the frequencies of these subsets were generally unchanged, with a modest and selective decrease in frequency and absolute number of CMPs in the absence of *PKCδ* (Fig 1E). Consistent with these observations, $PKC\delta^{-/-}$ BM cells showed a significantly increased frequency of *in vitro* colony-forming cells (CFU-C), measured at day 12 (Appendix Fig S1C). Furthermore, *in vivo* colony-forming unit-spleen (CFU-S) assays (Zhang *et al*, 2009), in which transplanted BM cells form hematopoietic colonies in recipient spleens, revealed equivalent numbers of colonies at day 8 after transplantation (CFU-S₈) and significantly higher numbers of colonies at day 13 (CFU-S₁₃) in recipients of KO BM (Appendix Fig S1D). The frequencies of mature circulating myeloid and erythroid cells were not detectably perturbed in PKCδ-deficient mice (Table EV1).

In both mice and humans, PKCδ deficiency causes increased B-cell proliferation and autoimmunity (Miyamoto *et al*, 2002; Kuehn *et al*, 2013). To determine whether this B-cell hyperproliferative phenotype due to PKCδ deficiency may be evident at the progenitor stage, we measured the frequency and absolute numbers of common lymphoid progenitors (CLP, Lin⁻IL-7Rα⁺Flt3⁺) and B lineage-committed progenitors (BLPs, Lin⁻IL7-Rα⁺Flt3⁺Ly-6D⁺). Our analyses indicated similar frequencies and absolute numbers of

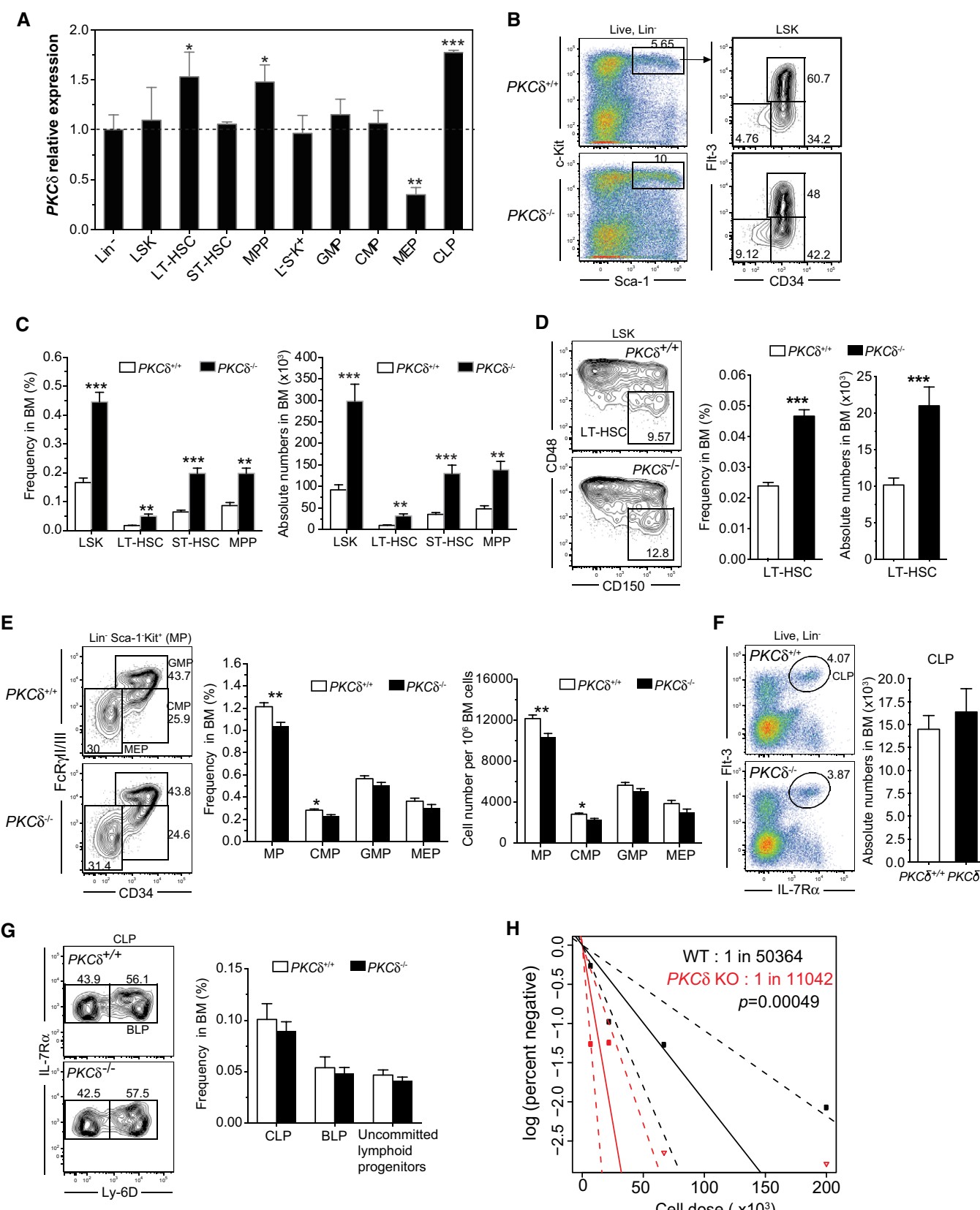

**Figure 1.**

**Figure 1.   PKCδ restricts HSPC pool size in the bone marrow.**

A    Quantitative real-time PCR analysis of *PKCδ* mRNA levels in FACS-sorted Lin⁻, LT-HSC, ST-HSC, MPP, L⁻S⁻K⁺, GMP, CMP, MEP, and CLP subsets from C56BL/6 wild-type (6- to 9-week-old) mice bone marrow. Levels of *PKCδ* expression were normalized to an internal control gene (β-actin). Expression of *PKCδ* is shown relative to Lineage negative (Lin⁻) cells whose expression was arbitrarily set to 1 ($n$ = 4 mice analyzed for each population).

B, C   (B) FACS plots depict the percentage of live, Lin⁻Sca1⁺c-Kit⁺ (LSK) (left panel), and primitive CD34⁻Flt3⁻, CD34⁺Flt3⁻, and CD34⁺Flt3⁺ HSPCs among LSK cells (right panel). (C) Bar charts show the average frequency (left) and absolute number (right) ± SEM of the indicated populations, analyzed from 2 femurs and 2 tibias per mouse ($n$ = 10 mice analyzed for each genotype).

D     FACS plots show percentages of "SLAM code" based stem and progenitor cells in the LSK population. Bar graphs show the average frequency (left) and absolute numbers (right) ± SEM of LT-HSC per 2 femurs and 2 tibias. ($n$ = 10 mice analyzed for each genotype).

E     FACS plots illustrate gating and frequencies of myeloid progenitors (MP), granulocyte–monocyte progenitor (GMP), common myeloid progenitor (CMP), and megakaryocyte–erythroid progenitor (MEP) population. Bar graphs show the frequencies (left) and absolute numbers (right) of L⁻S⁻K⁺ MP and GMP, CMP, and MEP populations in the BM of WT ($n$ = 12) and PKCδ KO ($n$ = 12).

F     FACS plots illustrate the gating and percentages of common lymphoid progenitors (CLP). Bar graph shows the absolute number of CLPs per 2 femurs and 2 tibias ($n$ = 5 mice for each genotype).

G     FACS plots illustrate the percentages of uncommitted lymphoid progenitors (Lin⁻ IL-7Rα⁺Flt3⁺Ly-6D⁻) and B lymphoid progenitors (BLP). Bar graph represents the average frequencies of CLPs, BLP, and uncommitted lymphoid progenitors in total BM ($n$ = 5 mice per genotype).

H     Limiting dilution analysis (LDA) demonstrates an increased frequency of long-term repopulating HSCs in PKCδ KO BM (solid red lines) compared with WT BM (black solid lines). Engraftment data shown at 14 weeks post-BMT. Plots depict the percentages of recipient mice containing < 1% CD45.2⁺ blood nucleated cells. Dotted lines represent the 95% confidence interval of the same ($P$ = 0.0005).

Data information: All data are presented as mean ± SEM. *$P$ < 0.05; **$P$ < 0.01, and ***$P$ < 0.001 by with one-way ANOVAs with Sidak's multiple comparisons test (A) or two-tailed Student's unpaired $t$-test (C-G) analysis for comparison of WT to PKCδ KO mice. Overall, test for differences in stem cell frequencies between WT to PKCδ KO mice was determined with likelihood ratio test of single-hit model (H).

CLPs and BLPs in the bone marrow of $PKC\delta^{+/+}$ and $PKC\delta^{-/-}$ mice (Fig 1F and G). Finally, to assess whether the increase in phenotypic HSCs in PKCδ-deficient mice correlated with an increase in hematopoietic reconstituting cells, we performed limit dilution transplantation assays, in which graded numbers of $PKC\delta^{+/+}$ or $PKC\delta^{-/-}$ BM cells (CD45.2⁺) were mixed with a fixed number ($2.5 \times 10^5$) of recipient type BM cells (CD45.1⁺) and transplanted into lethally irradiated CD45.1 recipients. These studies revealed a significant increase in reconstituting cell frequency in $PKC\delta^{-/-}$ as compared to $PKC\delta^{+/+}$ BM cells (1 in 11,042 versus 1 in 50,364, $P$ = 0.0005, Fig 1H). Thus, deficiency of PKCδ results in a significant immunophenotypic expansion of the LT-HSC, ST-HSC, and MPP populations, an increase in primitive CFU-S₁₃ activity, and an almost four-fold increase in the frequency of hematopoietic reconstituting cells as assayed by limit dilution engraftment studies. Collectively, these data suggest that functional PKCδ acts normally to limit expansion of the phenotypic and functional pool of HSPCs in adult bone marrow.

## Accelerated proliferation and reduced apoptosis of PKCδ-deficient HSPCs *in vivo*

We next investigated the underlying mechanisms that may account for the increased HSPC pool size in the BM of adult PKCδ-deficient mice. Because PKCδ has been implicated in regulating cell cycle and apoptosis in various cell lines (Basu & Pal, 2010), and loss of PKCδ is associated with a dramatic increase in HSPCs *in vivo* (Fig 1), we hypothesized that increased HSPC numbers in PKCδ-deficient BM could reflect an altered proliferation rate or decreased spontaneous cell death *in vivo*. To test this hypothesis, we examined possible changes in the cell cycle status of HSPCs during steady-state hematopoiesis using intracellular staining with Ki67/Hoechst. These analyses revealed a significant increase (two-fold) in the percentage of cycling (S/G2/M, identified as Ki67^hi Hoechst^hi) HSPCs and early progenitors in *PKCδ*-knockout mice as compared to WT controls (Fig 2A). PKCδ-KO HSCs further exhibited a substantial decrease in the quiescent G0 phase (Ki67^low Hoechst^low) and a concomitant

increase in the fraction of cells in the active G1 phase (Ki67⁺Hoechst^low) of cell cycle. Next, we performed a short-term (20 hr) *in vivo* BrdU labeling assay to quantify the frequency of actively proliferating cells in HSPC subsets (Fig 2B). In line with our findings using combinatorial Ki67/Hoechst staining, BrdU incorporation revealed an approximately 2.5-fold higher rate of BrdU incorporation in LT-HSCs from KO mice compared to *WT* controls (~20% versus 7.5%, Fig 2C). A moderate increase in BrdU⁺ cells was also observed in *PKCδ*-deficient ST-HSCs and MPPs, but not among myeloid progenitor stages (Fig 2B and C). These data suggest that loss of *PKCδ* activates cell cycle progression of primitive HSPCs, which in turn leads to their expansion.

We next assessed the frequencies of apoptotic cells among HSPC subsets using Annexin V/7-AAD staining. Percentages of Annexin V⁺/7-AAD⁻ (apoptotic) LT-HSCs were not substantially different between WT and $PKC\delta^{-/-}$ mice; however, ST-HSCs, MPPs, and myeloid progenitors from $PKC\delta^{-/-}$ mice exhibited a significantly lower rate (two-fold reduction) of spontaneous cell death as compared to their WT counterparts (Fig 2D). These data suggest that loss of PKCδ signaling in progenitor cells downstream of LT-HSCs exerts a protective role that reduces apoptotic rate. Collectively, these results indicate that the expansion of the primitive HSPC compartment apparent in PKCδ-deficient BM reflects alterations in both cell cycle progression and survival of HSPCs, with differential impact on distinct subsets of multi-potent and oligopotent progenitors.

## Loss of PKCδ enhances competitive repopulation capacity

The accumulation of primitive stem and progenitor cells in *PKCδ*-deficient mice BM could result from loss of *PKCδ* activity within HSPCs themselves or from defects in microenvironmental cues arising due to loss of *PKCδ* in hematopoietic or non-hematopoietic lineages that could indirectly affect their numbers. To distinguish hematopoietic system intrinsic versus extrinsic effects of PKCδ deficiency on HSPC function, we performed competitive BM transplants, in which total BM cells from WT or $PKC\delta^{-/-}$ mice were transplanted into lethally

**Figure 2. Accelerated proliferation and reduced apoptosis in subsets of PKCδ-deficient HSPCs.**

A   Representative FACS profiles of HSPC cell cycle analysis using combinatorial staining for Ki67 and Hoechst 33342. Bar charts depict the average percentage of cells in each phase of the cell cycle for each LSK subset from WT ($n = 6$) and PKCδ KO ($n = 7$) mice. Data compiled from two independent experiments.

B   FACS profiles of indicated BM subsets from WT and *PKCδ* KO mice 20 hr after BrdU injection.

C   Average percentages of cells in each phase of the cell cycle phases for each of the indicated HSPC subsets from WT and PKCδ KO mice. Data are pooled from two independent experiments (totaling $n = 6$–7 mice per genotype).

D   Representative FACS plots and summary of FACS data analyzing the frequency of apoptotic HSPC cells using co-staining for Annexin V and 7-AAD (WT and $PKCδ^{-/-}$) ($n = 7$ per genotype).

Data information: All data are presented as mean $\pm$ SEM, *$P < 0.05$ and **$P < 0.001$, with significance determined by two-tailed Student's unpaired $t$-test analysis.

irradiated congenic WT mice (CD45.1$^+$) along with equal numbers of recipient type (CD45.1$^+$) whole BM cells (Fig 3A). Analysis of peripheral blood chimerism in primary recipients revealed a slight but significant increase in engraftment of CD45.2$^+$ hematopoietic lineages, with particular enhancement of B-cell (B220$^+$) (Limnander *et al*, 2011) and myeloid (Gr1$^+$Mac-1$^+$) reconstitution, in mice receiving BM cells from $PKCδ^{-/-}$ as compared to control mice (Fig 3B).

To test the long-term self-renewal and differentiation capacities of $PKCδ^{-/-}$ HSPCs, we next conducted serial transplantation studies, which impose overt proliferative stress on HSCs (Min *et al*, 2008; Pietras *et al*, 2014; Rao *et al*, 2015). Defects in regulating self-renewal capacity or aberrant cell cycle activity can lead to HSC exhaustion in this setting. Total BM cells were harvested from primary recipients

20 weeks after initial transplantation and infused into lethally irradiated secondary recipients. Hematopoietic reconstitution was monitored by staining of peripheral blood cells sampled over 16 weeks after transplant (Fig 3A and B). Then, at 16 weeks after secondary transplant, total BM cells were harvested again to assess reconstitution potential in tertiary recipients (Fig 3A and B). Although accelerated cycling is frequently associated with progressive exhaustion of HSCs (Yilmaz *et al*, 2006; Min *et al*, 2008; Khandanpour *et al*, 2010; Pietras *et al*, 2011), mice receiving BM cells from $PKCδ^{-/-}$ mice showed enhanced reconstitution by $PKCδ^{-/-}$ cells in both the myeloid and B-cell compartments, as compared to control BM recipients, in both secondary and tertiary recipients (Fig 3B). Interestingly, $PKCδ^{-/-}$ T-cell reconstitution efficiency was enhanced only in tertiary, and not primary or secondary, recipients (Fig 3B). To assess

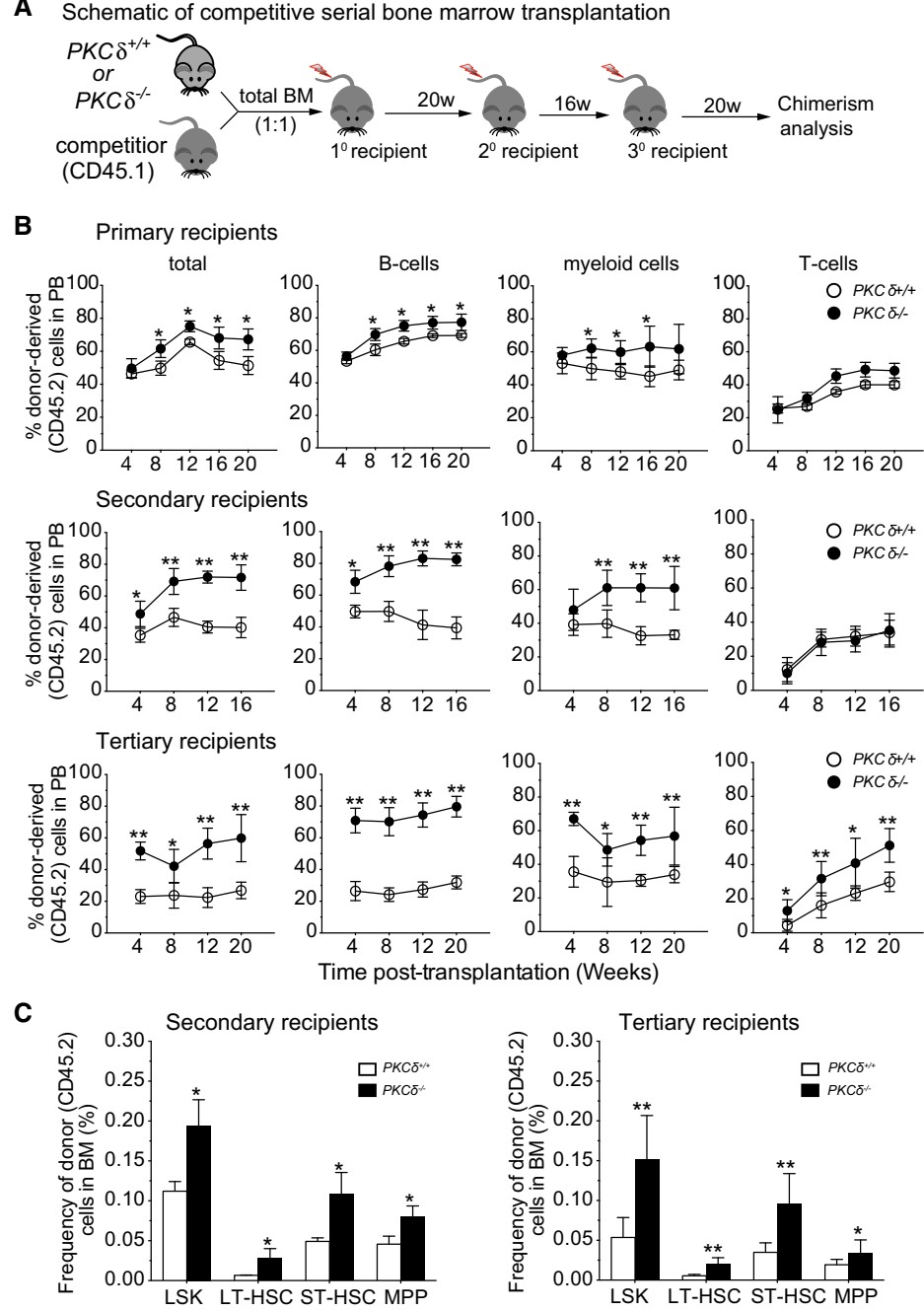

**Figure 3. Loss of PKCδ enhances competitive repopulation and self-renewal of HSPCs *in vivo* without exhaustion.**

A   Schematic of competitive BM transplantation assay.

B   Percent of total donor-derived, hematopoietic cells (CD45.2⁺), B cells (B220⁺), myeloid cells (CD11b⁺Gr1⁺), and T cells (CD3⁺) in the peripheral blood (PB) of recipient mice, as determined by FACS at the indicated time points. The statistical significance of differences was determined using two-way ANOVAs with Holm–Sidak's multiple comparisons tests ($n$ = 8 recipients per genotype in each experiment).

C   Percent donor-derived HSPCs in the BM of secondary (left) and tertiary (right) recipients of WT or *PKCδ⁻/⁻* marrow. Data are compiled from two independent experiments ($n$ = 8 recipients per genotype).

Data information: All data are presented as mean ± SD. *$P$ < 0.05, **$P$ < 0.001, with significance determined by repeated measures two-way ANOVA analysis with Sidak's multiple comparison tests for comparison of recipients of WT and PKCδ KO marrow at each time point and for each cell population.

whether the enhanced hematopoietic reconstitution potential of *PKCδ⁻/⁻* BM reflected the higher frequency of HSPCs in these animals, we measured the frequency of donor-derived (CD45.2⁺) HSPCs in secondary and tertiary recipients. Both the frequency and number of donor-derived primitive HSPCs were increased in recipients of PKCδ-deficient as compared to control marrow (Fig 3C).

Interestingly, competitive transplantation experiments using purified HSCs from $PKC\delta^{+/+}$ or $PKC\delta^{-/-}$ mice (Appendix Fig S1E) showed that, with transfer of equal cell numbers, $PKC\delta^{-/-}$ HSCs reconstitute irradiated recipients with equal efficiency compared to $PKC\delta^{+/+}$ HSCs (Appendix Fig S1F). Thus, on a per stem cell level, $PKC\delta^{-/-}$ HSCs are equipotent in reconstituting irradiated recipients, and the greater reconstitution levels seen with transfer of whole bone marrow likely reflects the higher numbers of primitive HSPCs in the marrow of $PKC\delta^{-/-}$ as compared to $PKC\delta^{+/+}$ mice. Altogether, these data indicate that, despite their increased cell cycle activity, PKCδ-deficient HSPCs are not inferior to wild-type HSPCs and support augmented long-term, multi-lineage hematopoietic reconstitution in bone marrow transplant and serial transplantation settings.

**Increase in immunophenotypic and functional HSPCs upon hematopoietic-specific deletion of PKCδ**

The competitive transplantation assays described above strongly suggest that the accumulation of primitive stem and progenitor cells in PKCδ-deficient mice is attributable to changes within the hematopoietic system itself. To eliminate potential effects of compensatory signals arising from loss of PKCδ in non-hematopoietic cells prior to hematopoietic cell transplantation, we generated a lineage-specific PKCδ knockout mouse by crossing $PKC\delta^{fl/fl}$ mice (Bezy et al, 2011) with mice carrying an Mx1-Cre transgene (Kuhn et al, 1995). This model allows for pan-hematopoietic interferon or polyinosine/polycytosine (pIpC)-inducible PKCδ allele excision and also drives allelic excision in a subset of stromal cells (Kuhn et al, 1995; Park et al, 2012). Administration of five doses of pIpC at 4–6 weeks after birth led to near complete absence of PKCδ protein in the Lin⁻ c-Kit⁺ subset of BM cells in $PKC\delta^{fl/fl}$;Mx-1Cre⁺ mice, as compared to $PKC\delta^{fl/fl}$; controls lacking the Mx-1Cre allele ($PKC\delta^{fl/fl}$;Mx-1Cre⁻ mice) (Fig EV2A). Moreover, consistent with previous reports evaluating germline-deficient PKCδ KO mice (Mecklenbrauker et al, 2002; Miyamoto et al, 2002; Limnander et al, 2011), we observed enlarged lymph nodes and a significantly expanded peripheral B220⁺ B-cell compartment at 24 weeks after pIpC induction in $PKC\delta^{fl/fl}$;Mx-1Cre⁺ mice (Fig EV2B). These PKCδ-deleted mice are hereafter referred to as $PKC\delta^{\Delta/\Delta}$ (cKO) and control mice, treated with pIpC but lacking the Mx-1Cre allele, are referred to as $PKC\delta^{fl/fl}$.

Analysis of $PKC\delta^{\Delta/\Delta}$ (cKO) and $PKC\delta^{fl/fl}$ mice at 4–8 weeks after the last pIpC injection revealed that acute deletion of PKCδ in hematopoietic and stromal lineages produced a significant increase in the frequency and number of Lin⁻ cells in the BM, but did not alter total BM cellularity or circulating mature blood lineages (Table EV2). These results contrast with observations in the full body PKCδ knockout mice, where bone marrow cellularity showed a slight, but consistent increase (Appendix Fig S1A), possibly reflecting subtle effects of PKCδ loss in non-hematopoietic cell types. Analysis of the Lin⁻ compartment in cKO BM revealed a 3- to 4-fold increase in the absolute number and frequency of primitive HSPCs as compared with $PKC\delta^{fl/fl}$ controls (Figs 4A and B, and EV2C), recapitulating results obtained with germline-deficient PKCδ⁻/⁻ BM (Fig 1B–D) and further confirming a role for PKCδ in regulation of HSPC pool size in adult BM.

Strikingly, prolonged (20–24 weeks after the last pIpC injection) hematopoietic and stromal-specific deletion of PKCδ further

increased (~5- to 10-fold) the number of primitive HSPCs, including LT- and ST-HSCs (Figs 4C and EV2C and D). Likewise, "SLAM code" based analysis (Kiel et al, 2005) of BM LSK cells showed an ~five-fold increase in LT-HSC (CD150⁺CD48⁻LSK) frequency in cKO mice as compared to $PKC\delta^{fl/fl}$ littermates (Fig EV2D). Together with the transplant data discussed above (Fig 3), these results suggest that increased HSPC pool size in PKCδ-deficient BM reflects loss of PKCδ activity within the hematopoietic system itself. Analysis of downstream lineage-committed CMP and CLP populations revealed that while acute deletion of PKCδ causes a slight but significant reduction in the numbers of these progenitors at 4–8 weeks post-induction, this reduction was not evident at later time points (20–24 weeks after PKCδ deletion; Fig EV2E and F). However, prolonged deletion of PKCδ did produce a significant increase in the frequency and number of early megakaryocyte (Pre-MegE and MkP) and erythroid progenitor subsets (Pre-CFU/E and CFU-E/Pro-E) in cKO mice BM, albeit at variable levels (Fig EV2G and H). Interestingly, this increase in Meg/E progenitors was not accompanied by an increase in the numbers of mature platelets or erythrocytes (Table EV2). We also observed a 2- to 3-fold increase in the frequency of LSK cells in the spleens of cKO mice (Fig EV3A); however, HSPC expansion in the BM was not accompanied by an increase in their abundance in the peripheral blood (Fig EV3B). Of note, despite dramatic expansion of the primitive HSPC pool, cKO mice did not show any signs of malignancy when followed up to 10 months after pIpC induction (data not shown). Collectively, these data suggest that hematopoietic deletion of PKCδ mirrors for the most part germline loss of this gene and leads to progressive accumulation of HSPCs in the BM with retention of relatively normal (with the exception of increased frequency of B-cell lineages) hematopoietic differentiation capacity.

To determine whether increased HSPC pool size in the BM of PKCδ cKO mice reflected changes in HSPC survival and proliferation rate, as in germline $PKC\delta^{-/-}$ mice (Figs 1–3), we measured the survival and proliferation of PKCδ cKO HSPCs using Annexin V and Ki67/Hoechst staining, respectively. HSPCs from cKO mice showed slightly lower rates of spontaneous apoptosis than HSPCs from $PKC\delta^{fl/fl}$ animals, particularly in the LSK, ST-HSC, and MPP compartments (Fig EV3C), confirming a pro-survival role for PKCδ in at least a subset of HSPCs. Cell cycle analysis by Ki67/Hoechst staining further revealed a significant increase in the percentage of HSPCs in G1 and S-G2-M phases of the cell cycle (Fig EV3D), and an increased rate of labeling with BrdU (Fig EV3E), in cKO as compared to control $PKC\delta^{fl/fl}$ mice. These results, in a distinct genetic model of induced PKCδ deficiency, provide further evidence that PKCδ functions to restrict HSPC pool size by modulating rates of HSPC proliferation and apoptosis.

We next investigated the functional capacity of HSPCs from PKCδ cKO BM using competitive transplantation assays. To discriminate potentially confounding effects of PKCδ deletion in stromal niche cells (Kuhn et al, 1995; Park et al, 2012) or PKCδ function during HSPC homing, we transplanted "undeleted" CD45.2⁺ BM cells from Mx1-Cre;$PKC\delta^{fl/fl}$ mice, or control $PKC\delta^{fl/fl}$ mice lacking Mx1-Cre, competitively (1:1 with CD45.1⁺ cells) into lethally irradiated CD45.1⁺ recipients (Appendix Fig S2A). At 8 weeks post-transplant, CD45.2⁺ donor cell engraftment levels ranged from 60 to 70% and were indistinguishable for recipients of Mx1-Cre;$PKC\delta^{fl/fl}$ versus $PKC\delta^{fl/fl}$ BM cells (Appendix Fig S2B). Animals in both groups were

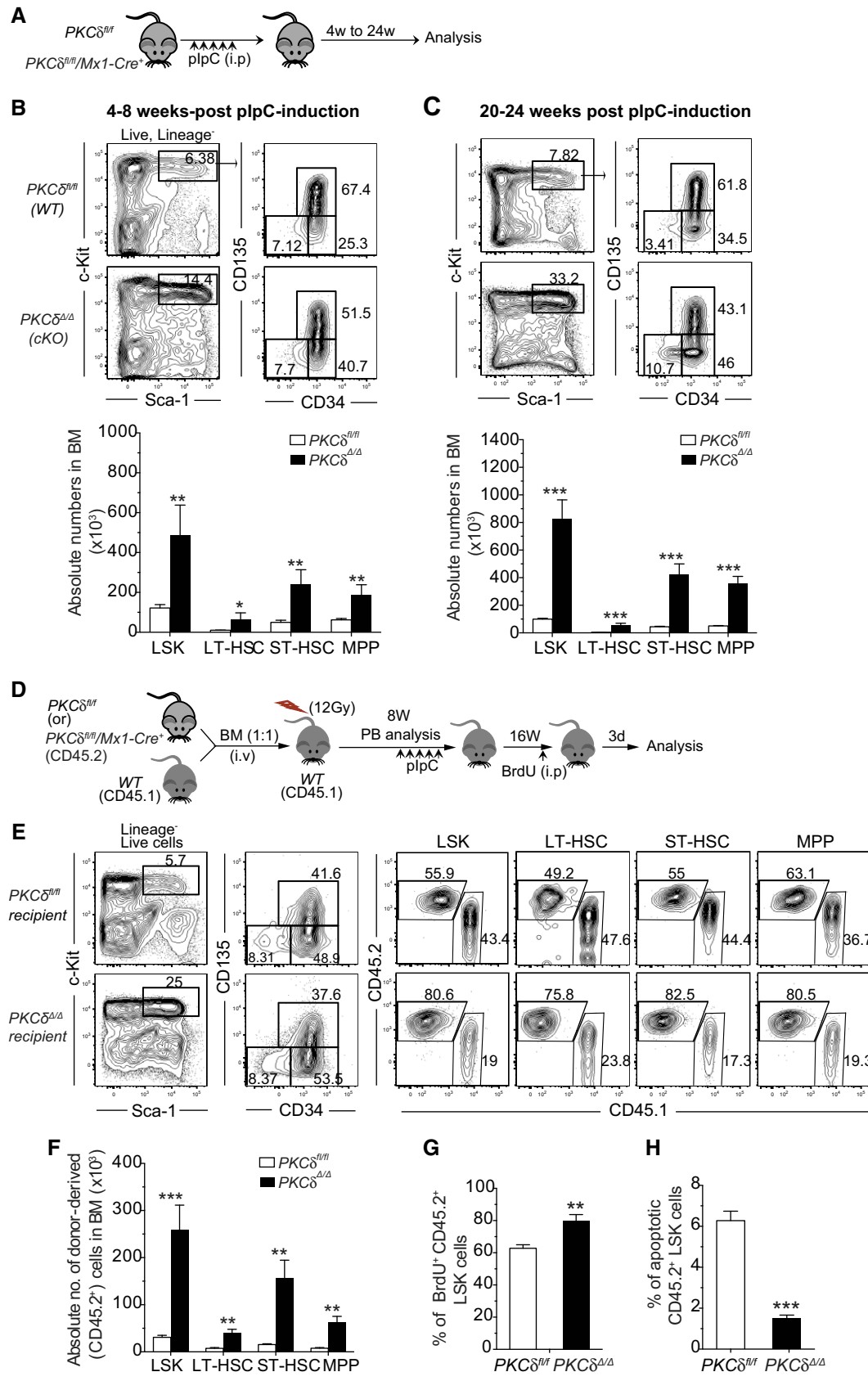

**Figure 4.**

**Figure 4.  Augmented HSPC pool size following loss of PKCδ in hematopoietic and stromal cells.**

A       Experimental design for pIpC treatment and analysis of *PKCδ* $^{fl/fl}$/*Mx-1Cre*$^-$ (*PKCδ*$^{fl/fl}$) and *PKCδ*$^{fl/fl}$/*Mx-1Cre*$^+$ (*PKCδ*$^{Δ/Δ}$, cKO) mice.

B, C    Representative FACS plots of *PKCδ*$^{fl/fl}$ and *cKO* BM 4–8 weeks (B) and 20–24 weeks (C) after pIpC treatment. Numbers in plot indicate the absolute numbers of the indicated subset within the gated population (*n* = 8–9 mice per genotype and time point).

D       Schematic of competitive BM transplants. Total BM cells from either *PKCδ*$^{fl/fl}$/*Mx-1Cre*$^-$ (control) or *PKCδ* $^{fl/fl}$/*Mx-1Cre*$^+$ mice were injected together with recipient type (CD45.1) BM cells (1:1) into lethally irradiated CD45.1 mice prior to pIpC treatment. Two months after transplant, both groups of mice received five doses of pIpC. Chimeras were analyzed at 16 weeks after the last pIpC injection.

E, F    (E) FACS plots show contribution of donor-derived (CD45.2$^+$) HSPCs in recipient BM (E). Calculated absolute numbers of donor-derived HSPC subsets in transplanted mice (F) (total of *n* = 7–8 mice per genotype).

G       Average percentage BrdU$^+$ HPSCs (3 days after BrdU injection into mice reconstituted 24 weeks previously with control or *cKO* marrow and injected 16 weeks previously with pIpC, see panel (D)) among donor-derived LSK cells in the control and cKO chimeras BM.

H       Percentages of apoptotic cells (Annexin V$^+$/7-AAD$^-$) in donor-derived LSK cells in control and cKO chimera mouse BM at 24 weeks after transplant and 16 weeks after pIpC injection. Data are compiled from two independent experiments (total of *n* = 7–8 mice per genotype).

Data information: Data presented as mean ± SEM. *$P$ < 0.05, **$P$ < 0.005, and ***$P$ < 0.001 by two-tailed Student's unpaired *t*-test analysis.

then treated with pIpC, which induced PKCδ deletion in the *Mx1-Cre;PKCδ*$^{fl/fl}$ recipients (Appendix Fig S2B). Subsequent analysis of peripheral blood (Appendix Fig S2) and BM HSPC chimerism (Fig 4D–H) at 8 and 16 weeks after pIpC induction revealed an increased representation of donor-derived CD45.2$^+$ peripheral blood leukocytes (Appendix Fig S2B) and BM HSPCs (Fig 4E and F) in recipients in which *PKCδ* was deleted as compared to recipients receiving control BM cells. *cKO* HSPCs also exhibited an increase in BrdU incorporation (Fig 4G) and a decreased in the percent apoptotic cells (Fig 4H), recapitulating the phenotype observed in germline *PKCδ* cKO mice and suggesting that *PKCδ* negatively regulates HSPC pool size in a hematopoietic cell-intrinsic manner, at least in part via effects on HSPC proliferation and survival.

**Enhanced recovery from myeloablation in PKCδ-deficient mice**

Collectively, our results in germline and inducible deletion models, coupled with analyses in competitive BM transplantation assays, strongly suggest that PKCδ negatively regulates HSC proliferation and survival during steady-state hematopoiesis and restrains hematopoietic regeneration in transplant settings. To test the *in vivo* consequences of *PKCδ* deficiency under distinct hematopoietic stress conditions, we next treated *cKO* or control *PKCδ*$^{fl/fl}$ mice with 5-fluorouracil (5-FU, 150 mg/kg; Fig 5A), a cytotoxic drug that initiates a BM stress response by selectively killing actively cycling stem and progenitor cells. 5-FU treatment causes a precipitous decline in the number of BM LSKs on days 2 through 7 (ablation phase) after treatment, which in turn leads to the reversible activation, rapid proliferation, and expansion of surviving quiescent HSCs from day 7 through 13 (recovery phase) (Schepers *et al*, 2012; Ehninger *et al*, 2014). When compared to control *PKCδ*$^{fl/fl}$ mice, PKCδ cKO mice displayed similar BM cell numbers and comparable kinetics of recovery of peripheral blood leukocytes, platelets, and red blood cells (RBC) after a single 5-FU treatment (Fig EV4A–C). Analysis of the BM HSPC compartment revealed similar depletion of LSK cells in *PKCδ*$^{fl/fl}$ and *cKO* mice at day 4 after 5-FU treatment (Fig 5B); however, *cKO* mice showed accelerated recovery, with a significant increase in HSPC frequency at 13d (recovery phase) and 28d (the stage at which homeostasis is re-established), as compared to control *PKCδ*$^{fl/fl}$ mice (Fig 5B and C). The spleen size was also increased in 5-FU-treated *cKO* mice as compared to controls at days 13 and 28 after 5-FU treatment (Fig EV4D), suggesting an enhancement of extra-medullary hematopoiesis. Finally, consistent with analyses of PKCδ$^{-/-}$ BM in the steady state, *PKCδ*-deficient HSPCs

in 5-FU-treated *cKO* mice displayed reduced apoptosis (Annexin V$^-$/7-AAD$^-$ cells; Fig 5D) and an enhanced BrdU incorporation rate (Fig 5E) during recovery from myelosuppression.

Defects in cell cycle regulation of HSCs resulting in unscheduled proliferation can cause precocious exhaustion after repetitive myeloablation (Schepers *et al*, 2012; Ehninger *et al*, 2014). Considering PKCδ as a potential negative regulator of HSC proliferation, we next tested whether stem cell exhaustion might occur after repetitive myeloablative stress. Control and cKO mice were treated weekly with 5-FU (150 mg/kg) for a total of 3 weeks and monitored for survival. Consistent with an increased fraction of proliferative HSCs, PKCδ cKO mice showed significantly enhanced sensitivity to repetitive 5-FU treatment, with a median survival of 19 days as compared to 31 days for control mice ($P$ = 0.014) (Fig 5F).

**Loss of PKCδ primes HSPC mitochondrial activity and energy metabolism**

Emerging studies highlight the importance of metabolic state as a critical determinant regulating stem cell number and fate (Ito *et al*, 2012; Yu *et al*, 2013; Burgess *et al*, 2014; Kohli & Passegue, 2014; Wang *et al*, 2014). In adult BM, primitive HSCs are thought to reside in hypoxic regions and rely on low ATP generating glycolytic metabolism to maintain their quiescence. However, HSCs can switch to mitochondrial oxidative phosphorylation (OXPHOS) in response to differentiation signals, infection, or stress to fulfill the increased energy demands associated with differentiation (Yu *et al*, 2013; Kohli & Passegue, 2014). Since *PKCδ* has been implicated as a major upstream regulator of metabolic signaling in may cell types and tissues (Bezy *et al*, 2011), we investigated whether *PKCδ* might regulate HSPC proliferation and survival by influencing HSPC energy metabolism. We first measured mitochondrial oxygen consumption rate (OCR), in FACS-purified HSPCs (LSK) using a Seahorse metabolic flux analyzer. *PKCδ*-deleted HSPCs showed significantly elevated basal OCR as compared to controls (Fig 6A and B). PKCδ *cKO* HSPCs also showed a significant increase in maximal OCR ($P$ = 0.001) following FCCP treatment, which contributed to a substantial increase in spare respiratory capacity (SRC) (Fig 6C), an indicator of cellular fitness to respond to an energetic demand that has been associated with increased survival during stress (Maryanovich *et al*, 2015). Notably, this increased OCR did not reflect a change in glycolysis, as measurements of basal extracellular acidification rate (ECAR) were similar, while HSPCs from PKCδ *cKO* mice had higher levels of maximal

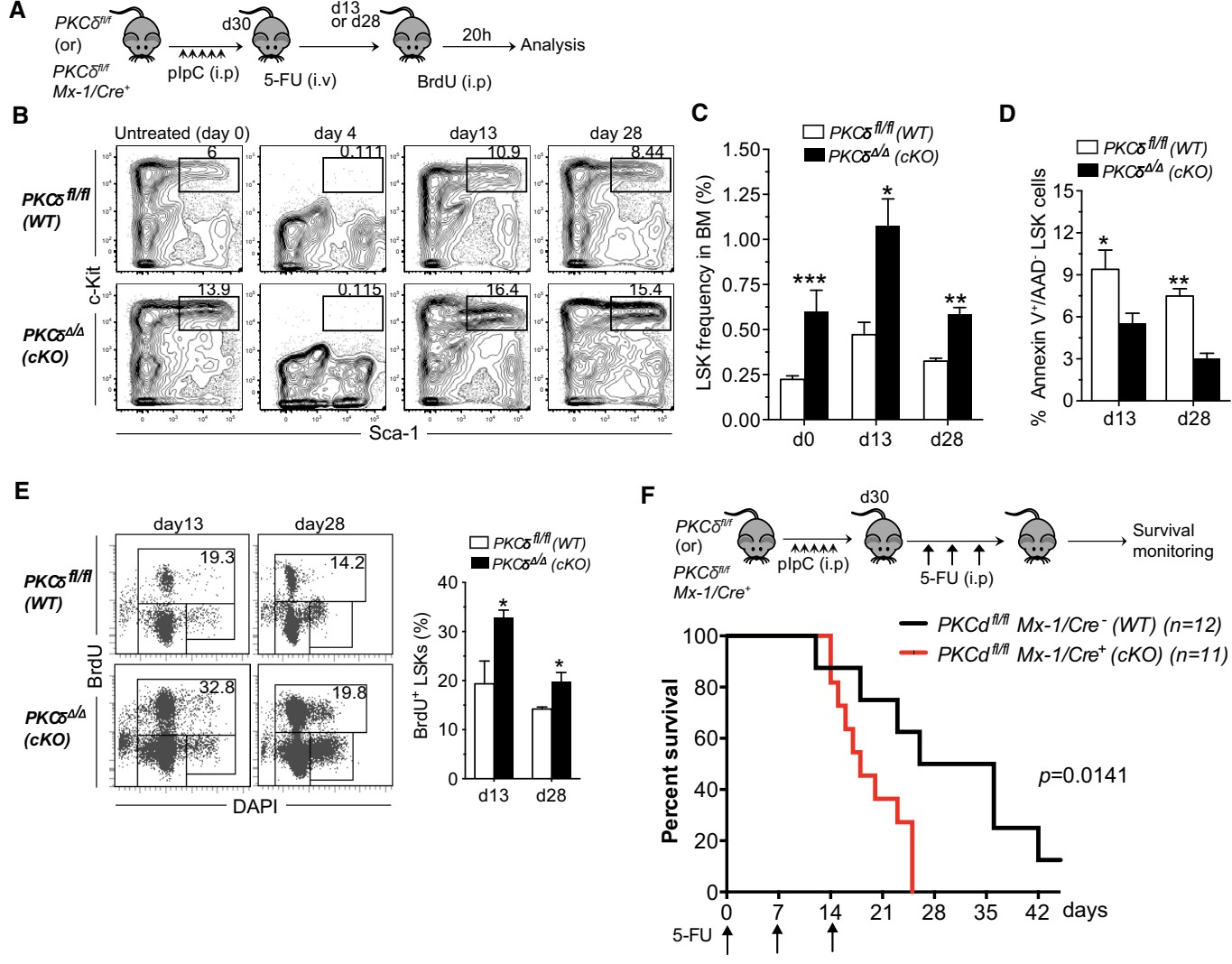

**Figure 5. Accelerated recovery in PKCδ-deficient mice after myeloablation by 5-FU.**

A  Experimental design. Thirty days after the last pIpC injection, *PKCδ*[fl/fl] and *PKCδ cKO* mice were treated with a single dose of 5-FU (150 mg/kg, i.p.) and analyzed subsequently by FACS and BrdU incorporation studies.

B  Kinetics of LSK cell recovery. Representative FACS profiles of BM LSK cells from WT and cKO mice at the indicated time points after 5-FU treatment. Numbers in the contour plots indicate the percentages of LSK subsets among Lin⁻ live cells.

C  Average absolute numbers of BM LSK cells at day 13 and day 28 after 5-FU treatment. Data are pooled from two independent experiments (total of n = 6–8 mice per time point and genotype).

D  Quantification of apoptotic HSPCs at the indicted time points after 5-FU treatment of control and *cKO* mice (n = 6 per genotype).

E  Representative FACS plots showing analysis of *in vivo* BrdU incorporation assays. Mice were received 2 mg of BrdU (i.p.) at d13 (left) or d28 (right) after 5-FU treatment. 20 hours later, BrdU incorporation in BM LSK cells was assessed using anti-BrdU antibodies. Bar chart at right indicates percent of BrdU⁺ LSK cells at d13 and d28 after 5-FU treatment (n = 6 per genotype and time point).

F  Kaplan–Meier survival cures of control (n = 12) and cKO (n = 11) mice after sequential 5-FU treatment. Mice were treated with 5-FU (150 mg/kg, i.p.) once each week for 3 weeks. Arrows indicate time points of 5-FU injection.

Data information: Data presented as mean ± SEM. The statistical significance was assessed using repeated measures two-way ANOVAs with Sidak's multiple comparisons tests (C) or by two-tailed Student's unpaired *t*-test analysis (D and E). *P < 0.05, **P < 0.01, and ***P < 0.001. Survival data (F) were analyzed using a log-rank nonparametric test (Mantel–Cox test) (P = 0.0141).

ECAR after challenge with Oligomycin, a mitochondrial ATP synthase inhibitor that forces cells to utilize glucose via glycolysis rather than mitochondrial OXPHOS to meet energy demands (Fig 6D and E). Moreover, the increased metabolic activity of *cKO* HSPCs was not attributable to increased glucose uptake, as demonstrated by similar levels of fluorescent glucose (2-NBDG) accumulation in HSPC subsets from *cKO* and control mice (Fig EV5A), further indicating that increased energy production is largely accounted for by augmented OXPHOS. PKCδ *cKO* ST-HSCs and MPPs also harbored significantly greater levels of ATP, although increased ATP levels were not observed in LT-HSCs (Fig 6F).

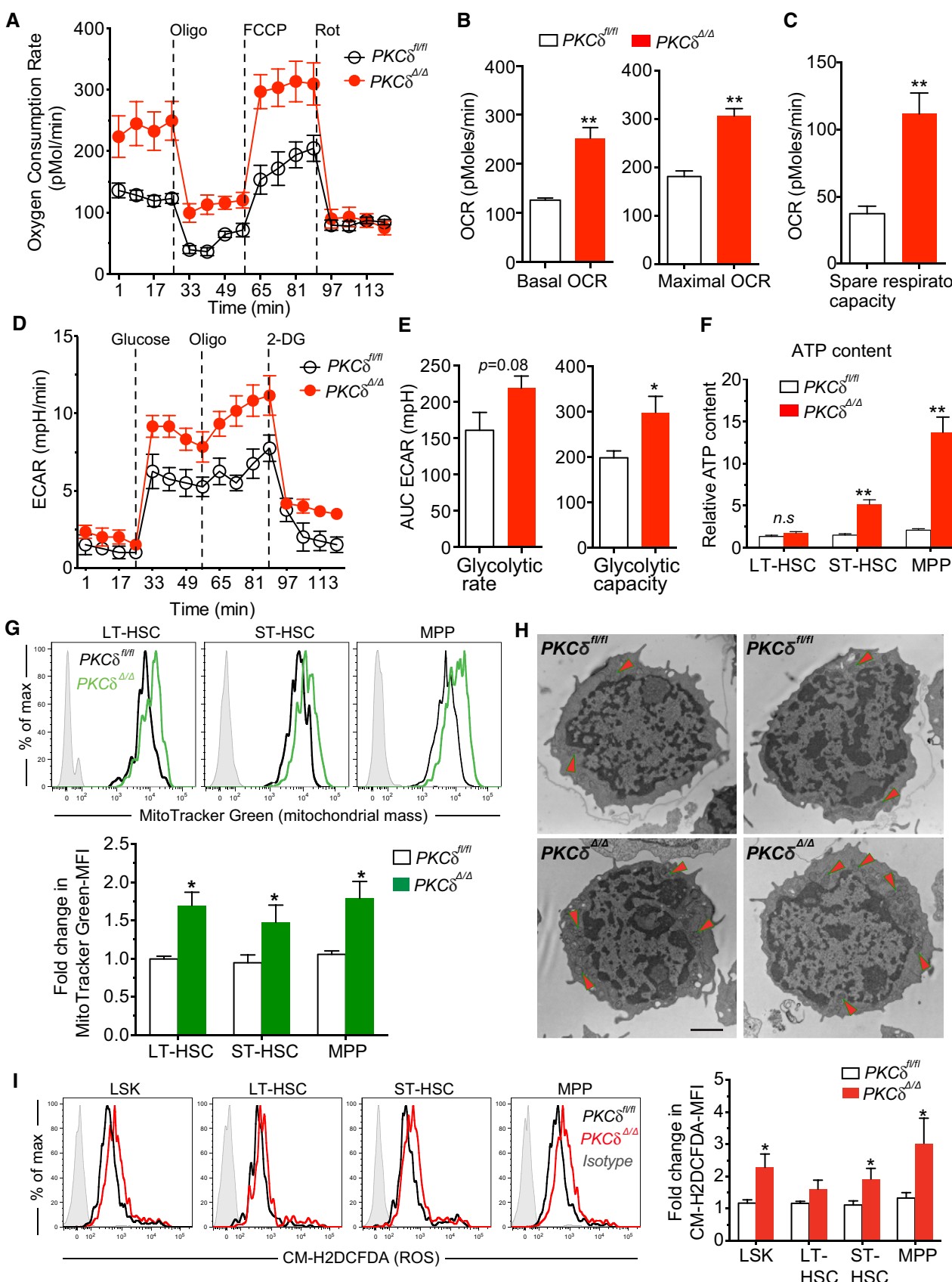

**Figure 6.**

◄

**Figure 6.  Increased mitochondrial number, activity, energy metabolism, and ROS levels in PKCδ-deleted HSPCs.**

A    MitoStress test revealed increased oxygen consumption rates (OCR) in *PKCδ*-deleted LSKs. BM LSK cells were sorted from WT and *PKCδ cKO* mice 12 weeks after pIpC treatment. OCR was measured at basal level and after sequential loading of ATP synthase inhibitor Oligomycin (350 nm), mitochondrial uncoupler, FCCP (10 μM), and electron transport chain inhibitor, Rotenone (1 μM) using Seahorse XF24 extracellular flux analyzer. Data are from three independent experiments (*n* = 9 mice). In each experiment, bone marrow cells from identical genotypes (*n* = 3 mice per genotype) were pooled and used for LSK cell sorting.

B, C   (B) Basal OCR was measured before inhibitors treatment (left); and the Maximal OCR capacity after FCCP treatment (right) (C). The spare respiratory capacity (SRC) of LSKs was calculated from the data shown in panel A (*n* = 9 per genotype, from three independent experiments).

D, E   (D) Glycolysis stress test. Extracellular acidification rates (ECAR) of LSKs were measured at basal conditions (unbuffered assay medium without glucose) and after sequential loading of glucose (7.5 mM), Oligomycin (350 nm), and 2-deoxyglucose (50 mM; 2DG, a glucose analog) (E). AUC of baseline ECAR levels (left), and glycolytic capacity (maximal ECAR) was calculated from the data shown in panel (D) (*n* = 7–8 per genotype, from three independent experiments).

F    Relative ATP content in indicated HSPC subsets determined using an ATP assay kit (*n* = 6 mice per genotype).

G    Representative FACS histograms of MitoTracker Green fluorescence intensity on indicated HSPC subsets from control and cKO mice. Relative MitoTracker Green mean fluorescence intensity (MFI) is quantified below for *n* = 6 mice of each genotype.

H    Representative electron microscopy images (4,800× magnification) of control and *PKCδ cKO* HSPCs. Arrows indicate the mitochondria-enriched regions (*n* = 3 mice per genotype). Scale bar represents 500 nm.

I    ROS levels in control and *PKCδ cKO* HSPCs as measured by FACS-based CM-H2DCFDA staining. Bar graph at right shows the relative CM-H2DCFDA MFI. Data compiled from three independent experiments (*n* = 6–8 per genotype).

Data information: The statistical significance of difference was assessed using two-tailed Student's unpaired *t*-test analysis. All data are presented as mean ± SEM, $*P < 0.05$, $**P < 0.001$.

Next, to assess whether the accelerated recovery of HSPCs in PKCδ *cKO* mice after 5-FU treatment (Fig 5B and C) was associated with increased metabolic activity, we measured mitochondrial oxygen consumption rate (OCR) of HSPCs from mice given a single 5-FU treatment 10 days prior (recovery phase). Consistent with analyses during steady state, HSPCs from myeloablated PKCδ *cKO* mice also showed a significant increase in both basal and maximal OCR, as well as ATP production compared to treated control cells (Appendix Fig S3A and B), suggesting that PKCδ sets a threshold for metabolic activation of HSPCs during steady-state and myeloablative regeneration.

To evaluate whether the increased mitochondrial function was associated with the changes in mitochondrial number and structure, we evaluated HSPC mitochondrial mass at 12–16 weeks after pIpC treatment using a mitochondrial membrane-specific fluorescent dye, MitoTracker Green. MitoTracker Green mean fluorescence was significantly higher in all primitive HSPC subsets from PKCδ *cKO* as compared to control subsets (Fig 6G). Ultrastructural analysis revealed a significant increase in number, but not size, of mitochondria in HSPCs purified from *cKO* as compared to control BM (Fig 6H). Consistent with this, mtDNA copy number was also markedly increased in PKCδ cKO HSPCs (not shown). Collectively, these results indicate that loss of *PKCδ* increases both mitochondrial number and mitochondrial activity, leading to metabolic priming and increased HSPC energy metabolism. As expected, this increased metabolic activity was accompanied by a moderate increase in reactive oxygen species (ROS) levels, a by-product of mitochondrial activity, in PKCδ-deleted *cKO* ST-HSCs and MPPs, but not in the LT-HSCs (Fig 6I). Although ROS can be an important endpoint signal for metabolic regulatory pathways implicated in HSC maintenance and function, the functional consequences of ROS-mediated actions are highly dose dependent. High ROS levels that exceed cellular antioxidant capacity can be detrimental to cell survival, while moderate increases in ROS may potentiate cell survival and proliferation (Ito *et al*, 2006; Ito & Suda, 2014). To test the relationship of increased ROS levels to the *in vivo* expansion of PKCδ-deleted HSPCs, we treated *cKO* or control mice with *N*-Acetyl-L-Cysteine (NAC), a potent ROS scavenger. NAC treatment was initiated after the last dose of poly (I:C) and continued for 8 weeks (Khandanpour

*et al*, 2010). NAC treatment reversed the increased ROS levels in PKCδ-deficient HSPC to normal levels (Fig EV5B), but had no effect on HSPC expansion (Fig EV5C), suggesting that reducing ROS levels alone is insufficient to rescue the augmented HSPC numbers in PKCδ-deficient mice. Collectively, these results identify PKCδ as a critical suppressor of HSPC oxidative metabolism and indicate that *PKCδ* deficiency can increase mitochondrial SRC, basal ATP, and ROS levels, thereby supporting the metabolic adaptation of HSPCs during physiological stress.

### PKCδ deficiency impacts multiple targets in signaling networks governing HSPC homeostasis

To gain further insights into the possible molecular mechanisms through which *PKCδ* restricts HSPC number and function in the steady state, we searched for downstream target genes that were dysregulated in the absence of *PKCδ*. Because PKCδ deficiency significantly altered HSPC cell cycle activity, survival, and metabolism, we focused on genes and pathways previously implicated in the regulation of these properties in HSPCs. Among these, *Bmi1*, a member of polycomb group (PcG) with transcriptional repressor activity, was significantly upregulated in PKCδ *cKO* HSPCs (Fig 7A). *Bmi1*-deficient adult mice display reduced HSC numbers, profound self-renewal defects, and impaired competitive repopulation capacity (Iwama *et al*, 2004). Conversely, *Bmi1* overexpression enhances HSPC number and self-renewal by favoring a shift toward more symmetric (versus asymmetric) stem cell divisions (Iwama *et al*, 2004). Our analyses of PKCδ-deficient *cKO* HSPCs at 12–16 weeks post-pIpC identified many phenotypic similarities with *Bmi1*-overexpressing HSPCs, including increased HSC frequency in adult but not in fetal stages, enhanced competitive repopulation capacity, and augmented self-renewal. Analysis of additional HSC transcriptional regulators whose loss of function has been associated with enhanced HSC self-renewal revealed that *Egr1* (Min *et al*, 2008; Gazit *et al*, 2013), *Klf4* (Heidel *et al*, 2013), and *Gfi1b* (Khandanpour *et al*, 2010) are all significantly down regulated in PKCδ *cKO* HSCs (Fig 7A).

We also examined the expression of lineage priming and differentiation markers in PKCδ-deficient and control HSPCs. Among

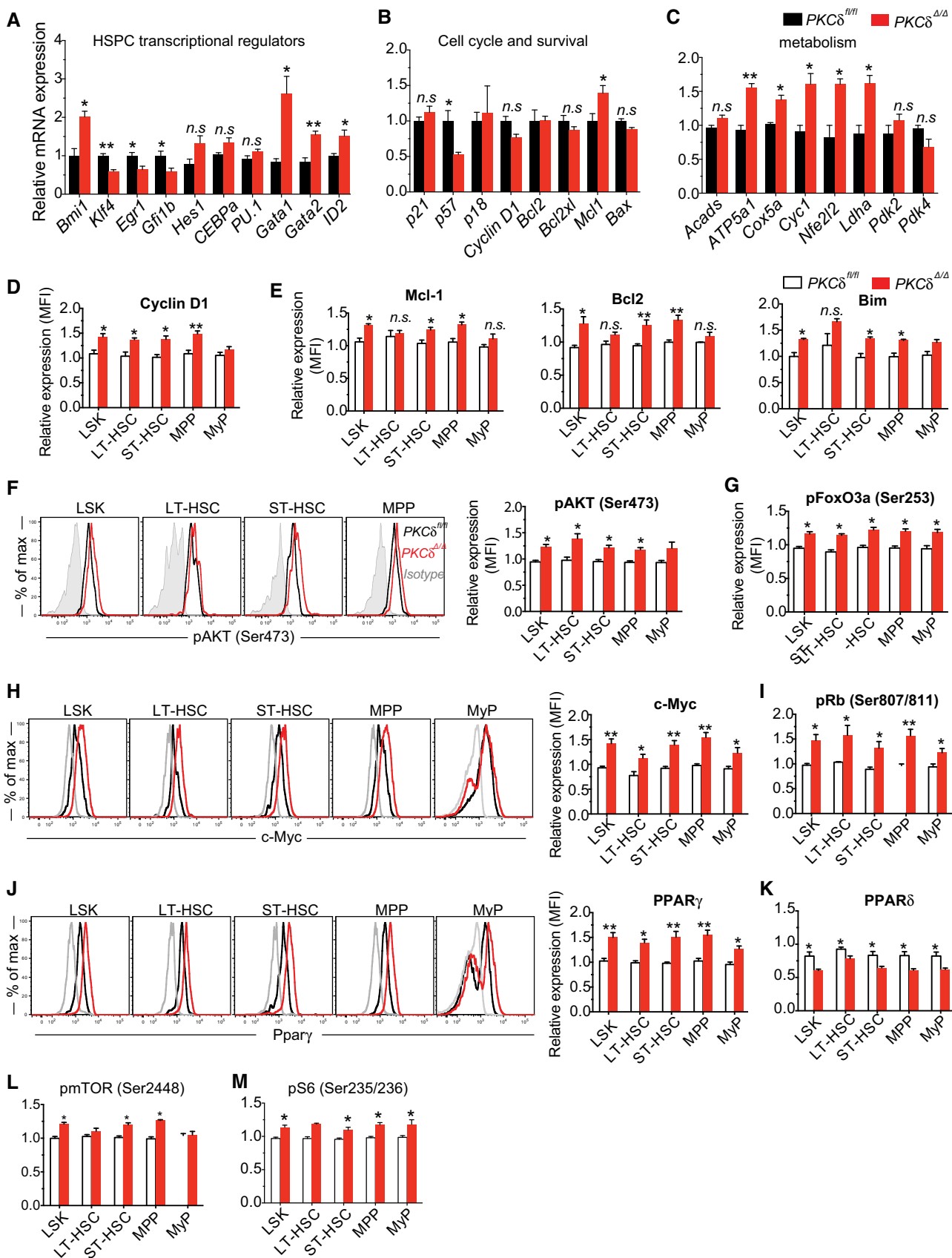

**Figure 7.**

**Figure 7.  PKCδ deficiency impacts multiple signaling cascade and transcriptional network governing HSPC homeostasis and function.**

A–C    Quantitative RT–PCR analysis of selected (A) HSPC transcriptional regulators, (B) cell cycle and apoptosis regulators, and (C) known HSPC metabolic regulators expression in bone marrow HSPCs ($n$ = 4 mice per genotype).
D–M    Intracellular flow cytometry analysis of Cyclin D1 protein (D); pro-survival regulators Mcl1, Blc2, and Bim protein (E); representative FACS histograms and bar graphs show relative levels of pAKT (Ser473) relative to WT controls (F); pFoxO3a (Ser253) (G); Total c-Myc protein (H); pRb (Ser807/811) (I); representative FACS histograms and bar graphs show total PPARγ protein levels relative to WT controls (J); Relative expression levels of total PPARδ protein in HSPCs (K); Activation of pmTOR (Ser2448) (L); pS6 (Ser235/236) and (Ser240/244) (M). Data shown in bar graphs in (D-M) are mean fluorescence intensity of indicated protein expression in *PKCδ cKO* HSPCs subsets relative to WT controls ($n$ = 4–6 mice per genotype per target).

Data information: The statistical significance of difference was assessed using two-tailed Student's unpaired *t*-test analysis for comparison of control and *PKCδ cKO* HSPCs. All data are presented as mean ± SEM, *$P$ < 0.05, **$P$ < 0.001.

the investigated genes, erythroid–megakaryocyte lineage regulators, *Gata1* and *Gata2* were significantly elevated in *cKO* HSPCs, whereas myeloid lineage regulators *Cebp1α* and *PU.1* (*Spi1*) showed equivalent levels of expression (Fig 7A). Despite unperturbed or even elevated differentiation potential, expression of *Id2,* a myeloid and B-cell differentiation factor and downstream target of β-catenin activation (Perry *et al*, 2011), was moderately upregulated in PKCδ *cKO* HSPCs, indicating possible activation of β-catenin in the absence of *PKCδ*. Analysis of cell cycle regulators revealed that expression of *p57* (*Cdkn1c*), previously implicated in HSC cell cycle regulation (Tesio & Trumpp, 2011), is significantly reduced in *PKCδ–*deleted *cKO* HSPCs as compared to control cells, whereas expression of *p18*, *p27* was unaltered (Fig 7B). Despite the significant increase in *Bmi1* transcript levels, its downstream target, *p16^{INK4A}*, was unaffected in HSPCs (Fig 7B). *PKCδ* deficiency also caused a significant increase in Cyclin D1 protein (Fig 7D), although the transcript level remained unchanged (Fig 7B). Among the cell survival regulators we explored, protein levels of the pro-survival mediators Mcl1, Bcl2, and Bim were moderately increased in PKCδ *cKO* HSPCs (Fig 7E), whereas protein levels of pPTEN (Ser380), pBad (Ser136), and PUMA were unperturbed (not shown).

Since loss of PKCδ significantly impacted the metabolic activity of *cKO* HSPCs, we also examined the critical regulators of these metabolic changes. Consistent with the increased OXPHOS and ATP content, expression of mitochondrial respiratory chain components, *ATP5A* (also called complex V), *Cox5a*, and *Cyc1*; expression of oxidative stress responsive gene *Nfe2l2* (*Nrf2*); and expression of *Ldha*, which catalyzes conversion of lactate to pyruvate to generate ATP in the mitochondrial electron transport chain, were all significantly elevated in PKCδ-deleted *cKO* HSCs (Fig 7C). No change in the expression of *Pdk2* and *Pdk4*, critical regulators of OXPHOS was noted. Nonetheless, this expression pattern fits well with the observed metabolic phenotypes and thus provides a molecular basis for increased metabolic activity of PKCδ-deficient HSCs.

PKCδ mediates cell context-dependent signal transduction pathways by phosphorylation of multiple downstream targets. We therefore assessed the effects of PKCδ deficiency on signal transduction in HSPCs. Because *PKCδ* acts upstream of PI3K/AKT, which transduces signals for HSPC survival, proliferation and differentiation through downstream effectors such as GSK3β, mTOR, NF-Κβ, MAPKs, and FoxO proteins (Tothova *et al*, 2007; Bezy *et al*, 2011; Buitenhuis, 2011; Perry *et al*, 2011; Lechman *et al*, 2012), we measured the individual components of these signaling cascades by intracellular flow cytometry. Total BM cells were starved in serum-free, cytokine-free media and stimulated with TPO and SCF. Activation of AKT, measured by elevated phosphorylation at Ser473, and its downstream target FoxO3a, measured by phosphorylation at

Ser253, was modestly increased in PKCδ-deleted *cKO* HSPCs as compared to controls (Fig 7F and G). Expression of c-Myc, a critical regulator of cell proliferation, mitochondrial biogenesis and metabolism, and phosphorylation of pRb (Ser807/811), a downstream regulator of Cyclin D1 implicated in the G1/S cell cycle transition and mitochondrial biogenesis were also significantly elevated in PKCδ-deficient *cKO* HSPCs (Fig 7H and I). We found no clear differences in the activation of MAPK signaling components, including p38 and Erk, or in activation of NF-κB signaling (as measured by phosphorylation of p65 at Ser536) in *PKCδ*-deleted HSPCs (not shown). Interestingly, PKCδ deficiency had an opposite effect on the expression of peroxisome proliferator-activated receptor isoforms (PPARs), whose function is implicated in HSPC maintenance (Ito *et al*, 2012). While PKCδ deficiency increased the expression of PPARγ, expression of PPARδ was significantly decreased (Fig 7J and K). Consistent with increased AKT activity, activation of mTOR (Ser2448) and its downstream target pS6 (Ser235/236) were also moderately increased (Fig 7L and M). Collectively, our findings suggest that augmented HSPC expansion and metabolic activity in PKCδ-deficient mice is a downstream consequence of dysregulation of multiple signaling pathways.

## Discussion

The expression and function of PKC isoforms are tissue and context-dependent (Basu & Pal, 2010; Bezy *et al*, 2011). While a recent study demonstrated that PKCζ/λ isoforms are dispensable for steady-state and regenerative hematopoiesis (Sengupta *et al*, 2011), our evaluation of PKCδ in HSPC homeostasis and hematopoietic regeneration reveals a critical role for this molecule in HSC proliferation, survival, and metabolic resiliency in both steady-state and stress hematopoiesis. Interestingly, despite the relatively broad pattern of *PKCδ* expression within the BM compartment, PKCδ-deficiency results in a rather specific impact on multi-potent HSPCs, leading to immunophenotypic expansion of LT-HSC, ST-HSC, and MPP populations. These data suggest that PKC isoforms play non-redundant and cell type-specific functions in regulating HSPC number and functional properties.

Chronic deficiency of PKCδ led to accelerated cell cycle activity and expansion of primitive HSPCs in adult mouse BM, but had little impact on downstream lineage-committed progenitors or mature cells. The availability and activity of different sets of PKCδ targets in stem and progenitor cells may contribute to the context-dependent phenotypes. Nonetheless, the observed HSPC expansion phenotype is clearly intrinsic to PKCδ function in the hematopoietic system, as demonstrated by the combined results from inducible deletion and competitive transplantation studies. Although both constitutive and

acute deletion of PKCδ in hematopoietic cells results in expansion of the primitive HSPC pool and a slight reduction in myeloid and lymphoid progenitor cell numbers, the reduction in myeloid progenitor numbers was reversed after chronic depletion of PKCδ in cKO mice. HSPC expansion in the BM was not accompanied by their increased abundance in peripheral blood as observed in other mouse models of HSPC hyperproliferation, including those lacking Egr1 or Gfi1b (Min et al, 2008; Khandanpour et al, 2010).

Exploring the potential mechanism(s) of HSPC expansion in PKCδ-deficient mice, we uncovered an enhanced metabolic activity, characterized by increased OXPHOS together with maintenance of glycolytic rate. PKCδ-deficient HSPCs also displayed elevated ROS levels; however, in vivo pharmacological treatment with NAC did not impede HSPC expansion in PKCδ-deficient marrow, suggesting that harmonizing ROS levels to the normal physiological range alone is not sufficient to prevent PKCδ-deficiency induced HSPC expansion. Remarkably, at the molecular level, rather than affecting a single regulator or pathway, PKCδ deficiency in HSPCs impacts multiple functionally important targets in interconnected pathways with many feedback and feed-forward inputs and outputs associated with cell cycle, survival, and metabolism. Extensive mRNA and protein-level expression profiling to quantify the total and activated signaling components implicated in HSPC biology identified AKT signaling and its downstream components as direct or indirect targets of PKCδ in HSPCs. Although PKCδ deficiency impacts each of these targets only moderately, it appears that this broad deregulation of HSPC signaling has significant functional consequences. We believe that the observed phenotypes are probably due to the cumulative effects of many of these dysregulated pathways. To our knowledge, our study provides the first demonstration that PKCδ acts as an upstream regulator of a network of HSPC regulatory genes, whose concerted deregulation in response to PKCδ loss suggests a previously unexplored connectivity among these disparate factors.

During hematopoietic homeostasis, a large proportion of HSCs reside in a metabolically and proliferatively quiescent state, and their unscheduled activation is detrimental to their long-term function (Ito et al, 2012; Yu et al, 2013; Burgess et al, 2014; Kohli & Passegue, 2014; Wang et al, 2014). These data support the general consensus in the field that active cell cycle state and metabolism are associated with HSC exhaustion leading to BM failure. Consistent with this, we found that PKCδ-deficient HSPCs exhibit accelerated failure in serial challenge with the myelosuppressive agent 5-FU. On the other hand, our findings in steady-state and transplant models indicate that deletion of PKCδ can augment HSPC pool size by promoting active proliferation while maintaining serial transplantation capacity. These data imply that, at least in certain conditions, HSC functional capacity and self-renewal is not compromised despite their active state. Similar observations have recently been reported with mice deficient in Egr1 (in which increased proliferation is accompanied by increased mobilization) (Min et al, 2008), p18INK4a (Yuan et al, 2004), cAbl (Rathinam et al, 2008), Itch1 (Rathinam et al, 2011), Gfi1b (Khandanpour et al, 2010), Llgl1 (Heidel et al, 2013), Dnmt3a (Challen et al, 2014), miR-126 (Lechman et al, 2012), and in Bmi1-overexpressing mice (Iwama et al, 2004) in which increased cell proliferation correlated with increased numbers of HSCs and self-renewal capacity. It remains to be determined whether an active metabolic state contributes to augmented HSPC numbers in these mice as well. Similarly, hyperactivation of PI3K/AKT signaling in

mouse models by inactivation of downstream negative regulators such as Pten, Pml, GSK3β (Yilmaz et al, 2006; Perry et al, 2011; Ito et al, 2012), and FoxO (Tothova et al, 2007) results in a transient increase in activated HSCs and has been reported to be accompanied by exhaustion, impaired repopulation capacity, or leukemia (Yilmaz et al, 2006). In contrast, PKCδ-deficient mice showed no signs of malignancy over a year follow-up.

A possible explanation for these unanticipated results is that although a significantly higher percentage of HSPCs exits G0, the absolute number of G0 HSPCs is increased by approximately 2–3-fold in PKCδ-deficient mice, due to the increased HSPC pool size in these animals and consistent with the observed increase in hematopoietic repopulating capacity in limit dilution studies. This observation is in line with the paradigm that long-term blood-forming potential resides predominantly in the G0 subset of HSPCs. Another possibility to reconcile these findings is that only a fraction of PKCδ-deficient HSPCs are metabolically active and cycling, perhaps related to their particular localization in the BM niche. It is also possible that HSC accumulation in PKCδ-deficient BM results from altered cell cycle division patterns, which may drive a fraction of HSCs to take more symmetric self-renewal than commitment divisions. In this context, it remains to be determined whether the loss of PKCδ induces a shift from glucose to lipid metabolism in HSPCs.

PKCδ-deletion leads to B-cell hyperproliferation and autoimmunity in mice (Mecklenbrauker et al, 2002; Miyamoto et al, 2002). Recently, human patients were identified with a homozygous loss-of-function mutation (c1840C>T, p.R614W) in PRKCD (PKCδ), resulting in deficient PKCδ expression (Kuehn et al, 2013). These patients presented with autoimmune lymphoproliferative syndrome (ALPS) characterized by lymphadenopathy, splenomegaly, and positive autoantibodies due to B-cell hyperproliferation, a phenotype mirroring the PKCδ knockout mice (Kuehn et al, 2013). Although we are not aware of any information suggesting a change in HSPC frequency or function in these patients, our data warrant investigation of such a possibility.

Follow-up studies will be needed to clarify the respective contributions of specific signaling outputs associated with the observed HSPC phenotypes in PKCδ-deficient mice to determine whether additional PKCδ targets may contribute to increased HSPC pool size and to assess whether loss of PKCδ also augments heterogeneity in myeloid lineage-committed progenitors (Paul et al, 2016). Furthermore, future studies should examine whether PKCδ exerts a similar influence on human hematopoietic stem and progenitors or on other tissue-specific adult stem cells. In summary, our findings reveal a previously unknown function of PKCδ in HSPC homeostasis and regeneration, and suggest that PKCδ may act as a rheostat for HSPC expansion. Reversible targeting of PKCδ, alone or in combination with previously known HSPC regulators, may be an attractive strategy for prolonging the maintenance of HSPCs ex vivo, and harnessing robust HSPC engraftment function in transplantation therapies.

## Materials and Methods

### Mice and treatment

Constitutive PKCδ knockout mice (Leitges et al, 2002) were back-crossed onto a C57BL/6J background for 14 generations (Bezy et al,

2011). To obtain homozygous *PKCδ* knockouts, we intercrossed *PKCδ*[+/−] mice. *PKCδ*[fl/fl] mice were described in Bezy *et al* (2011). Mice with *PKCδ*[fl/+] or *PKCδ*[fl/fl] alleles were crossed with *Mx1-Cre*[+] mice (Kuhn *et al*, 1995) to generate WT; Mx1-Cre; *PKCδ*[fl/fl]/*Mx-1Cre*[−] or *PKCδ*[fl/fl]/*Mx-1Cre*[+] (cKO) mice. Deletion of the *PKCδ* floxed allele was induced by injection of polyinosinic polycytidylic acid (pIpC; P1530, Sigma-Aldrich) at a dose of 500 μg per injection every other day for a total of five injections (i.p.) into 4- to 6-week-old mice. *PKCδ*[fl/fl] mice lacking the Cre transgene (*PKCδ*[fl/fl]/*Mx-1Cre*[−]) were treated in the same way and used as controls. To exclude possible effects of pIpC or interferons on HSCs, *PKCδ*[fl/fl]/*Mx-1Cre*[−] or *PKCδ*[fl/fl]/*Mx-1Cre*[+] mice were analyzed at least 4 weeks after the last pIpC injection (Khandanpour *et al*, 2010). Excision of the *PKCδ* allele was confirmed by Western blotting using total protein isolated from sorted Lin[−]Kit[+] BM cells. B6.SJLPtprca Pep3b/BoyJ (CD45.1) and *Mx-1/Cre* (B6-CD45.2) mice were purchased from The Jackson Laboratory and maintained at the Joslin Diabetes Center. All experiments involving mice were performed in accordance with the guidelines set by the Institutional Animal Care and Use Committees (IACUC) of Joslin Diabetes Center and Harvard University.

## Antibodies and FACS analysis

Total BM cells were harvested from long bones (two tibias and two femurs) by flushing with 25G needle using staining media (Dulbecco's PBS+ 5% FCS). Cells were filtered through 70 μm nylon mesh to obtain a single-cell suspension. Red blood cells were depleted by treatment with erythrocyte lysis buffer (ACK buffer, Lonza). To prevent non-specific binding of antibodies, cells were incubated with FcBlock (BD biosciences) or Rat-IgG (Sigma) for 20 min, before staining with fluorescent conjugated antibodies. The following monoclonal antibodies were used for FACS analysis and cell sorting: A mixture of biotinylated monoclonal antibodies CD3, CD4, CD8, CD19, B220, TER-119, Mac-1, and Gr-1 was used as the lineage mix (Lin). Anti-CD3e (17-A2), Anti-CD4 (L3T4), anti-CD8 (53-6.72), anti-B220 (RA3-6B2), anti-TER-119, Gr-1 (RB6-8C5), anti-Mac-1 (M1/70), APC-Kit (2B8), CD150 (TC15-12F12.2, all are from Biolegend); PE-Cy7- or PerCp5.5-anti-Sca-1 (E13-161.7), APC- or FITC-anti-CD34 (RAM34), anti-CD48 (HM48-1), PE-Cy7-CD127 (A7R34) (eBiosciences); PE-anti-Flt3 (A2F10.1), PE-FcγRII/III (2.4G2), FITC-ant-CD45.1, PE-Cy7-anti-CD45.2, FITC-anti-Ki-67 and FITC-Annexin V (BD Biosciences). SYTOX Blue (Invitrogen) was used to exclude dead cells during FACS analysis. Live, singlet cells were selected for gating and cell sorting. Cells were analyzed on a LSRII Flow Cytometer and sorted on a FACSAria-II cell sorter (BD biosciences). Data were analyzed either using FACS Diva software or FlowJo (version 9.4.4) software (Treestar). The absolute number of particular cell populations was calculated by multiplying the total number of live cells (trypan blue-negative) by the frequency of the indicated cell population determined by flow cytometry within the live cell (SYTOX Blue–negative) gate or by using cell-counting beads (BD).

## Bone marrow (BM) reconstitution assays

For limiting dilution long-term competitive transplantation assay, graded numbers ($7 \times 10^3$, $2.2 10^4$, $6.7 \times 10^4$, and $2 \times 10^5$) of total BM cells from *PKCδ*[+/+] or *PKCδ*[−/−] mice were sorted into individual wells of a 96-well plate containing fixed numbers ($2.5 \times 10^5$) of CD45.1[+] whole BM cells and transplanted intravenously into lethally irradiated CD45.1[+] mice. Sixteen weeks post-BMT, multi-lineage donor cell engraftment (CD45.2[+]) was determined by flow cytometry. A recipient mouse was considered positive if donor multi-lineage engraftment (CD45.2[+] blood nucleated cells) exceeded 1% in recipient peripheral blood. The frequency of functional repopulating units was calculated according to Poisson statistics using ELDA method (Hu & Smyth, 2009) (ELDA, http://bioinf.wehi.edu.au/software/elda/). For competitive transplantation assays, erythrocyte-depleted total bone marrow cells (BM) ($1 \times 10^6$) from 8- to 12-week-old *PKCδ*[+/+] or *PKCδ*[−/−] mice were mixed with $1 \times 10^6$ BM cells (1:1) from recipient type (competitor) and injected intravenously (in 200 μl PBS/mouse) into lethally irradiated (1000rads) CD45.1 (B6.SJLPtprca Pep3b/BoyJ) recipient mice. Hematopoietic reconstitution was assessed in recipient peripheral blood (PB) at the indicated times post-transplantation. For serial transplantation experiments, primary transplanted mice (described above) were sacrificed after 20 weeks, and $2 \times 10^6$ BM cells from each recipient mouse were injected into secondary preconditioned recipients (CD45.1). For HSC (LSKCD48[−]CD150[+]) competitive transplantation experiments, FACS-purified HSCs (200 cells) from WT or PKCδ KO mice (CD45.2[+]) were mixed with 1 million total bone marrow cells from WT mice (CD45.1[+]) and intravenously transplanted into lethally irradiated congenic WT recipient mice (CD45.1[+]). When using BM cells from conditional *PKCδ* knockout mice, total BM cells ($1 \times 10^6$) from *PKCδ*[fl/fl]/*Mx-1Cre*[−] (control) or *PKCδ*[fl/fl]/*Mx-1Cre*[+] mice were injected along with equal numbers of recipient type BM cells (CD45.1) into the lethally irradiated CD45.1 recipient mice. Recipient mice were allowed to reconstitute for 8 weeks, followed by pIpC injection (five doses) into control and *PKCδ*[fl/fl]/*Mx-1Cre*[+] mice, as described above. Donor-derived hematopoietic cells (CD45.2[+]) in PB and BM LSK cells were determined by FACS as described above.

## Colony-forming unit cell assay (CFU-C)

*In vitro* colony-forming activity was assessed using methylcellulose colony assay with Methocult GF3434 medium (StemCell Technologies). Briefly, freshly isolated BM cells were resuspended in 300 μl of IMDM and mixed in 3.3 ml of Methocult GF3434 medium. Cells were plated in 35 mm culture plate at a density of $1 \times 10^4$ BM cells/plate (1.1 ml/plate), in triplicate. On day 12, colonies containing at least 50 cells were scored under an inverted microscope. Differential colony count was performed on the basis of colony morphology, according to the StemCell Technologies criteria.

## Cell cycle analysis

For Ki-67 staining, cells were first stained with cells surface antibodies to identify the HSPCs as described above. Cells were then fixed in fixation buffer for 20 min at 4°C, washed twice with permeabilization buffer (BD), and then stained with anti-Ki67 antibody (BD biosciences) for 30 min at 4°C. Cells were washed once with permeabilization buffer and resuspended in FACS staining buffer containing 20 μg/ml Hoechst 33342 (Invitrogen). For short-term *in vivo* BrdU incorporation assay, mice were injected (i.p.) with 200 μl of BrdU/PBS solution (10 mg/ml; BrdU kit, BD bioscience). 20 hours

after injection, BM cells were harvested and stained with surface makers to identify HSPCs as indicated above. For long-term incorporation, BrdU was administered orally (1 mg/ml of BrdU in drinking water) for 7 days. Intracellular BrdU staining was performed according to the supplier's protocol (BD).

### Apoptosis assay

Erythrocyte-depleted BM cells ($1 \times 10^6$) were stained with antibodies to identify HSPC (Lin, c-Kit, Sca-1, Flt-3, and CD34) or myeloid progenitors (Lin, c-Kit, Sca-1, $F_c\gamma$RII/III, and CD34) as described above. The stained cells were resuspended in 100 μl of Annexin V binding buffer (BD) and incubated with Annexin V-V450 and 7-AAD (BD Pharmingen) for 15 min at room temperature. Cells were resuspended in additional 400 μl of Annexin V binding buffer and analyzed immediately using a BD LSR II flow cytometer.

### 5-FU treatment

5-Fluorouracil (5-FU) was administered to mice intraperitoneally at a dose of 150 mg/kg. Hematopoietic recovery was monitored by peripheral differential blood counts using Hemavet 950 (Drew Scientific) at indicated time points. For survival assay, 5-FU was administered intraperitoneally at a dose of 150 mg/kg weekly and survival was monitored daily. When using $PKC\delta^{fl/fl}/Mx\text{-}1^{Cre-}$ or $PKC\delta^{fl/fl}/Mx\text{-}1^{Cre+}$ mice, 5-FU was injected 30 days after the last pIpC injection.

### Metabolic assays

Real-time measurement of intact cellular oxygen consumption rate (OCR) and extracellular acidification rate (ECAR) was performed using the Seahorse XF24 extracellular flux analyzer (Seahorse Bioscience) as previously described (Yu *et al*, 2013; Maryanovich *et al*, 2015). Briefly, FACS-sorted LSK or MyP cells (4–5 replicate wells of $1 \times 10^5$ cells per well) were cultured 8 h in StemSpan serum-free culture medium (StemCell Technologies) supplemented with SCF (50 ng/ml) and TPO (50 ng/ml). Cells were harvested and plated in 24-well XF24 well plate (Seahorse Bioscience) pre-coated with BD Cell-Tak according to the suppliers instructions (BD). One hour prior to measurement, the medium was replaced with unbuffered assay medium, and then, plates were incubated at 37°C in a non-$CO_2$ incubator for 1 h for pH stabilization. For the MitoStress test, respiration was measured both at basal conditions and after sequential loading of ATP synthase inhibitor Oligomycin (350 nm), mitochondrial uncoupler carbonylcyanide 4-trifluoromethoxyphenylhydrazone (FCCP, 10 μM), and electron transport chain inhibitor, rotenone (1 μM) (Seahorse Bioscience). In the glycolysis stress test, respiration was measured both at basal conditions (unbuffered assay medium without glucose) and after sequential loading of glucose (7.5 mM), mitochondrial ATP synthase inhibitor Oligomycin (350 nm), and 2-deoxyglucose (50 mM; 2DG, a glucose analog, which inhibits glycolysis through competitive binding to glucose hexokinase) (Seahorse Bioscience).

For measurement of glucose uptake, BM cells were incubated at 37°C for the indicated times in StemSpan serum-free culture medium (StemCell Technologies) supplemented with SCF (50 ng/ml) and TPO (50 ng/ml) with or without 2-(n-(7-nitrobenz-2-oxa-1,3-diazol-4-yl)amino)-2-deoxyglucose (2-NBD glucose, 50 μM, Invitrogen). Cells were then washed once in HBSS and stained with surface markers to define stem and progenitor subsets as described above. Cells were analyzed by LSRII flow cytometer for 2-NBD glucose fluorescence in the FITC channel.

### ATP measurement

Intracellular ATP levels were determined using the ATP Bioluminescent somatic cell assay Kit (Sigma) following the manufacturer's instructions. Briefly, indicated populations were sorted directly into 96-well assay plate containing PBS (triplicates, 500 cells per well). Cells were incubated with the ATP releasing reagent, and luminescence was immediately quantified using Synergy Mx spectrometer (BioTeK).

### Determination of mitochondrial mass

For mitochondrial mass determination, total BM cells were incubated with 50 nM of MitoTracker Green (Invitrogen) at 37°C for 30 min after being stained with surface markers. Cells were then washed once with HBSS, and then, fluorescent intensity was measured by flow cytometry for MitoTracker Green in the FITC channel.

### Measurement of ROS levels

For measurement of cellular ROS levels, total BM cells were incubated with the redox-sensitive probe 2′,7′-dichlorodihydrofluorescein diacetate (CM-$H_2$DCFDA; Invitrogen) (5 μM) at 37°C for 20 min after staining with surface antibodies to define the stem and progenitor subsets as described above. Cells were then washed once with HBSS, resuspended in FACS staining buffer, and analyzed immediately by LSRII flow cytometer for CM-$H_2$DCFDA fluorescence in the FITC channel.

### NAC treatment

The antioxidant N-acetyl-L-cysteine (NAC; Sigma-Aldrich) was administered orally with the drinking water (10 mg/ml) for 8 weeks. Drinking water with freshly prepared NAC-supplemented water was replaced every second day. NAC treatment was started 1 week after the last pIpC treatment.

### *Ex vivo* HSC expansion

*Ex vivo* expansion of HSCs was performed as previously described elsewhere (Perry *et al*, 2011). Briefly, one hundred LT-HSCs (LSK CD48⁻CD34⁻CD150⁺) were sorted from WT mice BM and cultured in 96-well (U-bottom) in triplicate in StemSpan SFEM medium (StemCell Technologies) supplemented with mSCF (20 ng/ml) and mTPO (20 ng/ml). Cells were treated with or without Mallotoxin (MTX, previously called Rottlerin, 5 μM) or Indolactam V (10 μM) (Abcam) for indicated time. One-half of the medium was replaced every 3 days. At the indicated days of culture, cells were harvested and analyzed to determine the percent of cells retaining a Lin⁻Sca-1⁺c-Kit⁺ phenotype by FACS.

## Total and phosphoproteome analysis by intracellular FACS staining

Erythrocyte-depleted BM cells were stained with fluorescent conjugated antibodies to identify the HSPC subsets as described above. For measurement of intracellular phosphoproteins and total proteins, cells were starved for 2 h in StemSpan serum-free medium without cytokines and then stimulated with mSCF and mTPO (20 ng/ml) for 10 min. Cells were then fixed with BD Cytofix buffer for 20 min at 4°C, washed once with 1× Perm/Wash buffer (BD). For measurement of intracellular total protein expression of indicated cell cycle and apoptotic regulators, cells were stained with indicated antibodies without prior starvation and cytokine stimulation. Cells were incubated with Cytopermeabilization buffer Plus for 10 min on ice. After a single wash with 1× Perm/Wash buffer, cells were re-fixed with BD Cytofix/Cytoperm buffer for 10 min on ice. Cells were washed and incubated with the indicated primary and secondary antibodies for 1 h at room temperature. Cells were analyzed on a LSRII Flow Cytometer. Complete list of antibodies used were listed in Appendix Table S2.

## Western blotting

Total protein lysate from the indicated cell types was subjected to standard Western blotting method. Following antibodies were used: rabbit anti-PKCδ (2058, Cell Signaling Technology) and β-actin (Santa Cruz).

## Electron microscopy

FACS-sorted BM HSPCs (LSKCD135⁻) were pooled from identical genotypes ($10^5$ cells per genotype). Cells were fixed overnight in 2.5% glutaraldehyde at 4°C. Cells were pelleted at 3,000 $g$ for 10 min at 4°C. Cells were then submitted to Harvard Medical School Electron Microscope Facility for standard transmission electron microscopy ultrastructural analyses. Processed cells were imaged with AMT camera system with direct magnification of 4,800×. Scale bar represents 500 nm. Data are representative of 50 scanned cells per genotype.

## Gene expression analysis

Total RNA was isolated from the indicated cell types using Trizol or RNeasy Microkit (QIAGEN), and treated with RNase-free DNase. cDNA was generated with the SuperScript III First-Strand Synthesis System for RT–PCR (Invitrogen). qRT–PCR transcript levels were determined relative to *β-actin*. Transcript levels were determined using Taqman probes (FAM) specific for indicated genes. All reactions were performed in duplicate. Complete list of Taqman gene expression probes used are listed in Appendix Table S1.

## Statistical analysis

Statistical analysis was performed with the use of two-tailed Student's unpaired *t*-test analysis (when the statistical significance of differences between two groups was assessed) or one-way ANOVAs with subsequent Bonferroni posttest (when the statistical significance of differences between more than two groups was assessed), or two-way ANOVAs with subsequent Holm–Sidak's multiple comparison tests with alpha 0.05 as significant (when comparing between groups; for long-term reconstitution assays and for hematopoietic recovery analysis) with Prism software version 6.0 (GraphPad Inc). For the Kaplan–Meier analysis of survival curves, a log-rank nonparametric test (Mantel-Cox test) was performed. Limiting dilution analysis (LDA) was performed with ELDA (http://bioinfo.wehi.edu.au/software/elda/). Significance is denoted with asterisks (*$P < 0.05$, **$P < 0.01$, ***$P < 0.001$).

**Expanded View** for this article is available online.

## Acknowledgements

We thank A. Wakabayashi, G. Buruzula, and J. LaVecchio at Joslin DRC/HSCI Flow Cytometry Core (NIH P30DK036836) for excellent flow cytometry support, T. Serwold for FITC-Ly6D antibody, M.E. Patti for total and phosphoprotein antibodies, metabolic assay reagents, and technical advice on Seahorse and metabolic assays, D. Pober for statistical review of the manuscript, and all the members of Wagers, Kulkarni, and Kahn laboratories for helpful discussions. This work was supported by grants from the HHS | National Institutes of Health (NIH) (R01HL088582, DP2OD004345, PN2EY018244, and HL135287) to AJW. RNK acknowledges support from R01 DK67536 and R01 DK103215.

## Author contributions

TNR: conceived, designed and performed experiments, analyzed and assembled the data, and co-wrote the manuscript with AJW. MKG: performed RT–PCR and Western blot experiments. YCJ, SS and TT helped with Seahorse experiments. OB, SS, and CRK provided *PKCδ* KO mice, *PKCδ^{fl/fl}* mice and reagents, and edited manuscript. LDW: provided preliminary data, assisted with experiments, and edited the manuscript. RNK edited the manuscript. AJW: conceived and designed experiments, provided advice, and co-wrote the manuscript.

## Conflict of interest

AJW is a scientific advisor for Frequency Therapeutics, and a co-founder and scientific advisor for Elevian, Inc. LDW, is consultant and owns equity in Magenta Therapeutics. BO is an employee of Pfizer Inc. None of these commercial entities were involved in any manner in the conceptualization, design, data collection, analysis, decision to publish, or preparation of the manuscript. All authors declare no conflicts of interest.

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
