## [Review Process File · The EMBO Journal]

Attenuation of PKC δ enhances metabolic activity and promotes expansion of blood progenitors

Tata Nageswara Rao, Manoj K Gupta, Samir Softic, Leo D Wang, Young C Jang, Thomas Thomou, Olivier Bezy, Rohit N Kulkarni, C. Ronald Kahn, and Amy J. Wagers

Review timeline:	Submission date:	6th Aug 2018
	Editorial Decision:	17th Aug 2018
	Revision received:	31st Aug 2018
	Editorial Decision:	7th Sep 2018
	Revision received:	8th Sep 2018
	Accepted:	12th Sep 2018

Editor: Daniel Klimmeck

Transaction Report:

(Note: An earlier version of this manuscript was assessed by another journal and was then transferred to The EMBO Journal. As the original review of the manuscript was performed outside of The EMBO Journal's transparent review process policy, this Peer Review Process information is not available here. With the exception of the correction of typographical or spelling errors that could be a source of ambiguity, letters and reports are not edited. The original formatting of letters and referee reports may not be reflected in this compilation.)

1st Editorial Decision

17th Aug 2018

Thank you again for the submission of your manuscript (EMBOJ-2018-100409) to The EMBO Journal. We have carefully assessed your manuscript and the point-by-point response provided to the referee concerns that were raised during re-review at a different journal. In addition, and as mentioned before, we decided to involve an arbitrating expert to evaluate the revised version of your work, with respect to technical robustness, conceptual advance and overall suitability of your work for publication in The EMBO Journal.

As you will see from the report provided below, the arbitrating advisor states the robustness of your work as well as the overall interest and value of your results for the community and s/he thus is supportive of publication at The EMBO Journal.

Based on the positive expert's view together with our own assessment, we conclude that the previous referees' concerns regarding more detailed exploration of the mechanism downstream of PKC- δ in HSPCs and the relevance of this function in additional contexts do not need to be further addressed for publication at The EMBO Journal.

Thus, we decided to proceed with publication of your work at The EMBO Journal pending minor issues related to formatting and data representation as outlined below are conclusively addressed.

Once we have received the revised version, we should then be able to swiftly proceed with formal acceptance and production of the manuscript.

ARBITRATING ADVISOR'S REPORT:

'The study clearly represents a huge amount of work, with rigorous conditional deletion and serial transplantation studies. I'm not concerned about the increase in HSC proliferation. It's true that increased self-renewal tends to be associated with increased quiescence. However, there are two documented exceptions, e.g. NRas increases the proliferation of a subset of HSCs while increasing overall self-renewal potential. There are also a few examples of genetic modifications that seem to increase proliferation and self-renewal of the overall pool. These examples are particularly interesting. Not much is known yet about HSC metabolism.'

Corresponding Author Name: Amy J. Wagers

Journal Submitted to: The EMBO journal

Manuscript Number: EMBOJ-2018-100409R